# A Matrix Variational Auto-Encoder for Variant Effect Prediction in Pharmacogenes

## Abstract

Variant effect predictors (VEPs) are designed to predict the impact of protein variants on cellular function, traditionally using data from multiple sequence alignments (MSAs). This assumes that natural variants are fit, a premise challenged by pharmacogenomics, where some pharmacogenes have low evolutionary pressure. In this context, deep mutational scanning (DMS) datasets are of particular interest since they provide quantitative fitness scores for variants. In this work, we propose a transformer-based matrix variational auto-encoder architecture and evaluate its performances on 33 DMS datasets corresponding to 26 drug target and absorption-distribution-metabolism-excretion (ADME) proteins available in the ProteinGym benchmark. Our model trained on MSAs (matVAE-MSA) outperforms a model similar to the widely used VEPs in pharmacogenomics, and sets a new zero-shot prediction benchmark for 2 proteins related to the Noonan syndrome. We compare matVAE-MSA with matENC-DMS, a model with similar capacity, but trained on DMS data in a 5-fold supervised cross-validation framework. matENC-DMS outperforms matVAE-MSA for 15 out of 33 DMS datasets, including all ADME, and certain drug target proteins. Although our models do not outperform the best baseline models, our results help shed new light on the role of evolutionary pressure for the validity of the premise of VEP design. In turn motivating the development of DMS datasets to improve VEPs on pharmacogene-related proteins.

## 1 Introduction

Variant effect predictors (VEPs) are mathematical models aiming at predicting the effect of one or multiple variants in a sequence of amino-acids (AAs). The effect of a protein variant is typically defined as a loss or gain of function of a cell carrying the variant, compared to a cell carrying a wild-type (WT) protein without variant. The accurate prediction of variant effect has many promising applications for personalized medicine, particularly in the field of pharmacogenomics, where variants on drug targets or Absorption-Distribution-Metabolism-Excretion (ADME) proteins are of particular interest (Huang et al., 2016). In this context, VEPs can be used to assess individual patient response to chemotherapeutic treatments from their genetic background, thus eliminating the need for multiple attempts at treatments. The most effective VEPs have been designed using data from multiple sequence alignments (MSAs) and based on the *conservation assumption*: fit variants were selected out by nature and thus, learning a distribution over variants found in nature implicitly captures the biochemical constraints that characterize fit variants. New sequencing techniques combined with machine learning could lead to significant advances in variant effect prediction, by providing quantitative data in region of the protein sequence space unexplored in existing MSA datasets. Deep mutational scanning (DMS) has recently emerged as a way to yield large-scale datasets of protein quantitative fitness scores (Fowler & Fields, 2014). The fitness scores can also be obtained with different selection assays, allowing to quantify various effects, e.g. effect on phenotype or effect on structure. DMS thus allows to challenge the conservation assumption of VEPs design from MSAs. This is in turn is of particular importance in pharmacogenomics, since pharmacogenes are generally under low evolutionary pressure (Zhou et al., 2022; Ingelman-Sundberg et al., 2018).

In this article, we design a VEP and use it to evaluate the validity of the conservation assumption for pharmacogene-related proteins. Our architecture exploits the structure of variational auto-encoders (VAEs) and allows models of similar capacity and designs to be trained on both MSA and DMS

data. We also exploit a transformer architecture in order to improve upon existing VAE-based VEPs. Both VAE and transformers are key components of the best performing models in the ProteinGym benchmark (Notin et al., 2023a). We experiment with a VAE-based model exploiting multimodal priors, and we derive a matrix encoding scheme inspired from linear matrix decomposition to replace the input flattening operation found for instance in DeepSequence.

**Contributions** Our contributions are summarized as follows:

1. We design protein specific models combining a VAE and a transformer for variant effect prediction. We study their zero-shot prediction performances on 33 deep mutational scanning (DMS) datasets of drug related and ADME proteins available in the ProteinGym benchmark.

2. We study the impact on performances of using expressive latent prior distributions when the models are trained on MSA data available in ProteinGym. We experiment with standard mixture of Gaussian (MOG) and VampPrior.

3. We adapt our models to directly predict labels from DMS datasets using a prediction head from the latent space, thus preserving our model capacity. In light of the comparison in performances of the models trained unsupervised on MSA and supervised DMS label data, we discuss the extent of the validity of the conservation assumption.

## 1.1 RELATED WORKS

**Zero-shot predictors** VEPs exploiting site-independent position-wise frequencies of AAs in MSAs remain the methods of choice in pharmacology, e.g. SIFT or Polyphen-2 (Ng & Henikoff, 2003; Adzhubei et al., 2010; Durbin, 1998). However, other models can achieve much better zero-shot prediction performances on at least one pharmacogene-related protein DMS dataset (Details in Table A.6), according to the recent ProteinGym variant effect prediction benchmark (Notin et al., 2023a). Many of these models compute the functional cellular effect of a variant $v$ compared to a wild-type sequence $\underline{\mathbf{x}}^{(wt)}$, via the log-likelihood ratio:

$$\hat{y} = \ln \frac{p(\underline{\mathbf{x}}^{(v)})}{p(\underline{\mathbf{x}}^{(wt)})}, \tag{1}$$

where $p(.)$ is a generative probability density chosen to maximize $p(\underline{\mathbf{x}}^{(v)})$, for sequences $\underline{\mathbf{x}}^{(v)}$ from the MSA. For instance, the Evolutionary Scale modeling (ESM) approaches (Rives et al., 2021; Lin et al., 2023), rely solely on a transformer-based protein language model (PLM) for modeling the distribution over sequences in MSAs. Tranception (Notin et al., 2022a) additionally integrates predictions using position-wise frequencies of AAs in MSAs. TranceptEVE (Notin et al., 2022b) combines the Tranception model with a VAE-based model (Frazer et al., 2021) for AA sequence modeling. Other methods such as Masked Inverse Folding (MIF) (Yang et al., 2023) learn to predict protein sequences from a given structure. VESPA (Marquet et al., 2022) combines protein sequence embedding from PLMs with known bio-mechanical properties of AAs to predict variant effect with a linear regression model. Other model do not rely on the ratio in (1) to compute variant effect. MSA Transformer (Rao et al., 2021) is based on ESM and uses axial attention to optimize a masking loss over an entire MSA, rather than on individual sequences. It learns a representation of Hamming distances in the MSA and the hamming distance to WT sequence is used as a proxy for variant effect. GEMME (Laine et al., 2019) predicts variant effect via the distance to WT sequence in an evolutionary tree. This approach shows very good performances and has several order of magnitude fewer parameters than transformer-based approaches. DeepSequence (Riesselman et al., 2018) introduces a VAE and approximates the distribution of input data $\underline{\mathbf{x}}$ (Eq. (1)) with the variational evidence lower bound.

**Supervised learning predictors** Recently, several models combining DMS and MSA datasets have been proposed Hsu et al. (2022). The general idea is to combine sequence embeddings, e.g. sequence one-hot encoding, with evolutionary fitness scores from pretrained models such as ESM or DeepSequence. ProteinNPT is a conditional pseudo-generative model designed for exploiting DMS data, jointly with MSA data in a semi-supervised framework (Notin et al., 2023b). In addition to their novel architecture, the authors introduce several baselines consisting in exploiting prediction

scores from zero-shot prediction models pretrained on MSA, including DeepSequence and MSA Transformer. SPIRED is a recent framework able to predict fitness scores as well as protein structure (Chen et al., 2024). A pretrained ESM model is used for sequence embedding, and graph attention networks and multilayer perceptron are trained using DMS data in a supervised framework.

**Multi-modal prior distributions for VAEs** VAEs, e.g. DeepSequence, assume that the input data $\underline{\mathbf{x}} \in \{0, 1\}^{L \times d}$ are generated from a latent variable of a $D$-dimensional vector space: $\mathbf{z} \in \mathbb{R}^D$. The latent variable is assumed drawn from a Gaussian prior $p(\mathbf{z})$, and the generative process is modeled with a distribution $p_\theta(\underline{\mathbf{x}}|\mathbf{z})$. The explicit modeling of the latent variable $\mathbf{z}$ is an interesting feature of VAEs, because it allows to put a formal prior distribution on the latent space. The other mentioned models do not impose such structure, although interestingly the learnt representations in ESM was shown to correlate with known bio-mechanical properties of AAs (Rives et al., 2021). Multimodal mixture of Gaussian (MOG) priors have been proposed as latent prior distributions for unsupervised clustering tasks (Dilokthanakul et al., 2017) with VAE. The authors used trainable mean and covariances in latent space and showed through data sampling that the learnt mixture components corresponded to meaningful characteristics of the input data. This was shown to have potential implications for model interpretability in biological contexts (Varolgüneş et al., 2020). Further, the VampPrior has been designed so that the statistics of the mixture components explicitly depend on input space prototypes Tomczak & Welling (2018). This provides meaningful variables to probe for interpretability rather than using a sampling scheme. To the best of the authors knowledge, current methods employing VAEs for VEP have only been designed with unimodal prior distributions.

## 2 METHODS

A detailed description of the matVAE-MSA architecture is provided in section 2.1. A reduction of the architecture with similar capacity and that can be trained on DMS data: matENC-DMS, is proposed in section 2.2. The datasets that are used to train and evaluate the models are introduced in section A.1.

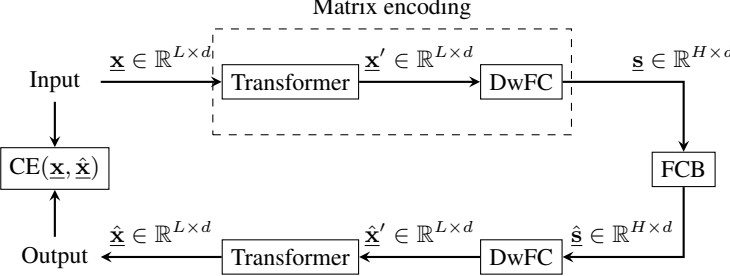

Figure 1: Model architecture of matVAE-MSA. DwFC: Dimensionwise fully connected layer. CE: Cross-entropy loss. FCB: Fully connected bottleneck.

### 2.1 MODEL DESCRIPTION

#### 2.1.1 MATRIX DECOMPOSITION AND ENCODING

Matrix decomposition is a set of methods in linear algebra consisting in decomposing a matrix $\underline{\mathbf{x}} \in \mathbb{R}^{L \times d}$ (here $d < L$), into two (or more) matrices with interesting structure, e.g. unitary, triangular, diagonal. For instance, the QR decomposition of $\underline{\mathbf{x}} \in \mathbb{R}^{L \times d}$ is of the form $\underline{\mathbf{x}} = \mathbf{Q}[\mathbf{R}^T, \mathbf{0}^T]^T$, where $\mathbf{Q} \in \mathbb{R}^{L \times L}$ is unitary, $\mathbf{R} \in \mathbb{R}^{d \times d}$ is upper triangular and $\mathbf{0}$ is a $(L - d \times d)$-dimensional matrix of zeros. We call matrix encoding the use of the low dimension factor, here the $\mathbf{R}$ matrix, as a compressed representation of the input $\underline{\mathbf{x}}$. For QR decomposition, the matrix encoding can be formulated:

$$[\underline{\mathbf{s}}^T, \mathbf{0}^T]^T = \mathbf{W}\underline{\mathbf{x}} \in \mathbb{R}^{L \times d}, \tag{2}$$

where $\mathbf{W} = \mathbf{Q}^{-1} \in \mathbb{R}^{L \times L}$ is a linear transform and $\underline{\mathbf{s}} = \mathbf{R}$. Linear decomposition methods are often related to singular value decomposition, for instance the diagonal elements of $\mathbf{R}$ in the QR

decomposition are the singular values of $\underline{\mathbf{x}}$. However, for one-hot encoded sequences of AAs, the singular values are counts of each AA in the sequence, and the singular vectors are a permutation of $\mathbb{I}_d$ determined by the ordering of the counts of the AAs. In other words, the encoding produced with such linear methods does not account for the global or relative position of AAs in the sequence, and in particular, randomly permuting the rows of $\underline{\mathbf{x}}$ leads to the same encoding. To ensure that the model is flexible enough to learn a useful encoding, we propose to learn a representation of $\underline{\mathbf{x}}$ with a transformer, prior to reducing the first dimension to a fixed $H < L$ with a trainable linear transform. This is formulated as follows:

$$\underline{\mathbf{x}}' = \text{Transformer}(\underline{\mathbf{x}}) \in \mathbb{R}^{L \times d},$$

$$\underline{\mathbf{s}} = \text{DwFC}(\underline{\mathbf{x}}') \in \mathbb{R}^{H \times d},$$

where $\text{Transformer}(.)$ and $\text{DwFC}(.)$ are specified in the paragraphs below.

**Transformer**  Transformers are effective sequence models that can transfer information between any two positions within a sequence. The model we use is similar to the multi-layer encoding transformer in (Vaswani et al., 2017). Individual layers encode an input sequence $\underline{\mathbf{x}} \in \mathbb{R}^{L \times d}$ into a sequence $\underline{\mathbf{x}}' \in \mathbb{R}^{L \times d}$ as follows:

$$
\begin{aligned}
\underline{\mathbf{x}}_1 &= \text{Norm}(\tau_1 \underline{\mathbf{x}} + (1 - \tau_1)\text{Attn}(\underline{\mathbf{x}})), \\
\underline{\mathbf{x}}' &= \text{Norm}(\tau' \underline{\mathbf{x}}_1 + (1 - \tau')\text{FC}(\underline{\mathbf{x}}_1)),
\end{aligned}
\tag{3}
$$

where $\text{Norm}(.)$ is a Layer Normalization (Ba et al., 2016), $\text{FC}(.)$ is a fully connected (FC) network with ReLU activations, and $\text{Attn}(.)$ is the masked scaled dot product attention from (Vaswani et al., 2017). We use trainable $\tau_1, \tau' \in [0, 1]^2$ to control the gradient flow in the gated skip connections (He et al., 2016). Note that we do not use positional encoding since (Rives et al., 2021) showed that PLMs did not necessarily benefit from it. Instead, structure information is encoded in the mask of the attention layer (See section A.2).

**Dimension-wise FC (DwFC)**  We call DwFC the FC linear layer inspired from Eq. 2. This layer is similar to a flattening followed by a linear transform with bias, but requires less parameters since the same linear transform is used across dimensions. This operation replaces the direct flattening of the input in DeepSequence. $\underline{\mathbf{x}}' \in \mathbb{R}^{L \times d}$ is encoded in a protein length independent representation $\underline{\mathbf{s}} \in \mathbb{R}^{H \times d}$ as follows:

$$\underline{\mathbf{s}} = \mathbf{U}\underline{\mathbf{x}}' + \mathbf{b}, \tag{4}$$

where $\mathbf{U} \in \mathbb{R}^{H \times L}$ and $\mathbf{b} \in \mathbb{R}^H$ are trainable weight and bias parameters.

### 2.1.2   FULLY CONNECTED BOTTLENECK (FCB)

The FCB is similar to a classical VAE (Kingma & Welling, 2014). $\underline{\mathbf{s}} \in \mathbb{R}^{H \times d}$ is flattened and then encoded into a latent representation of fixed dimension:

$$\mathbf{h} = \text{FC}(\text{Vec}(\underline{\mathbf{s}})) \in \mathbb{R}^D, \tag{5}$$

where $\text{Vec}(.)$ denotes the flattening operation. $\mathbf{h}$ is then used to compute the statistics of the latent vector $\mathbf{z} \in \mathbb{R}^D$:

$$q_\phi(\mathbf{z}|\underline{\mathbf{x}}) = \mathcal{N}\left(\mathbf{z}; f_\mu(\mathbf{h}), f_\sigma(\mathbf{h})\right), \tag{6}$$

where $\mathbf{h}$ is a function of $\underline{\mathbf{x}}$ and $f_\mu$, respectively $f_\sigma$, is a 1-layer fully connected network returning a mean vector, respectively a diagonal covariance matrix. For training, we introduce robustness by drawing the latent vector $\mathbf{z} \sim q_\phi(\mathbf{z}|\underline{\mathbf{x}})$ using the reparameterization trick. At test time, we use $\mathbf{z} = f_\mu(\mathbf{h})$ to reduce stochasticity. The latent distribution $q_\phi(\mathbf{z}|\underline{\mathbf{x}})$ is learnt to be close to a prior distribution $p(\mathbf{z})$ with respect to the Kullback-Leibler divergence (KLD). The latent vector $\mathbf{z} \in \mathbb{R}^D$ is then used as input to a decoder network that aims to reconstruct the input $\underline{\mathbf{s}}$. The output to the decoder, and thus of the FCB, is denoted $\hat{\underline{\mathbf{s}}} \in \mathbb{R}^{H \times d}$.

**Mixture of Gaussian (MOG) Prior**  To extend the work carried out in DeepSequence, we choose prior distributions formulated as a MOG:

$$p(\mathbf{z}) = \frac{1}{M} \sum_{k=1}^{M} p_k(\mathbf{z}; \boldsymbol{\mu}_k, \boldsymbol{\Sigma}_k), \tag{7}$$

where for $k = 1, \cdots, M$, $p_k(\mathbf{z}; \boldsymbol{\mu}_k, \boldsymbol{\Sigma}_k) = \mathcal{N}(\mathbf{z}; \boldsymbol{\mu}_k, \boldsymbol{\Sigma}_k)$ are multivariate Normal distributions with diagonal covariance $\boldsymbol{\Sigma}_k \in \mathbb{R}^{D \times D}$ and mean $\boldsymbol{\mu}_k \in \mathbb{R}^D$. As opposed to Gaussian distribution, MOG are multimodal and add more structure in latent space. This in turn can lead to a more expressive generative model, with a latent space able to important differences in input space in different modes, and fully use individual modes to encode subtle differences.

**VampPrior** The VAMP prior (Tomczak & Welling, 2018) is a special case of MOG with means and covariance that are functions of trainable prototypes in input space. That is, for $k = 1, \ldots, M$, prototypes $\underline{\mathbf{u}}_k \in \{0, 1\}^{L \times d}$ are used to compute

$$\mathbf{h}_k = \text{FC} \left( \text{DwFC} \left( \text{Transformer} \left( \underline{\mathbf{u}}_k \right) \right) \right),$$

where DwFC(.) and Transformer(.) are defined in 2.1.1, and FC(.) is the fully connected encoding defined in (5). The mean and covariance of each mixture components are then computed with $f_\mu$ and $f_\sigma$, similarly to 6. The prior distribution is finally expressed as:

$$p(\mathbf{z}) = \frac{1}{M} \sum_{k=1}^{M} \mathcal{N}(\mathbf{z}; f_\mu(\mathbf{h}_k), f_\sigma(\mathbf{h}_k)),$$

where for $k = 1, \ldots, M$, $\mathbf{h}_k$ depends upon the $k$-th trainable prototype $\underline{\mathbf{u}}_k$. In addition to the potential benefits of multi-modal priors mentioned before, the VampPrior could provide a way to interpret the modes of the prior distribution, in light of prototypes in input space.

### 2.1.3 DECODING

The decoding process denoted $p_\theta(\underline{\mathbf{x}}|\mathbf{z})$, is symmetrical to the encoder, i.e. consists of a decoding FC layer, and a dimension wise FC layer followed by a transformer. One important difference is that the decoding transformer includes a temperature softmax output operation to ensure that the rows of the reconstructed $\hat{\underline{\mathbf{x}}} \in \mathbb{R}^{L \times d}$ define proper discrete distributions.

### 2.1.4 LOSS FUNCTION

Our loss function is directly derived from the negative evidence lower bound (ELBO) used in VAEs (Kingma & Welling, 2014). In the ELBO, one term corresponds to an expected reconstruction loss (defined in section A.3). The other term is the KLD between the approximated posterior distribution $q_\phi$, a Gaussian, and the prior $p$, a mixture of Gaussian distributions. The KLD between $q_\phi$ and $p$ has no closed form expression and is therefore approximated with an upper bound (Durrieu et al., 2012):

$$D_{KL} \left( q \,||\, \sum_{k=1}^{M} w_k p_k \right) \leq -\ln \sum_{k=1}^{M} w_k e^{-D_{KL}(q||p_k)}, \tag{8}$$

where $q, p_1, \ldots, p_M$ are distributions such that $q$ is absolutely continuous with respect to $p_1, \ldots, p_M$, for $k = 1, \ldots, M$ $w_k \geq 0$ and $\sum_{k=1}^{M} w_k = 1$. A proof of the inequality can be found in (Rodríguez Gálvez, 2024, Appendix 6.B). Here, $q = q_\phi(\mathbf{z}|\underline{\mathbf{x}})$, for $k = 1, \ldots, M$ $w_k = \frac{1}{M}$ and $p_k$ are mixture components defined in (7). The negative ELBO loss function is finally written using (8) as follows:

$$l(\underline{\mathbf{x}}; \theta, \phi) = - \left( \ln \frac{1}{M} \sum_{k=1}^{M} e^{-D_{KL}(q_\phi||p_k)} + \mathbb{E}_{q_\phi(\mathbf{z}|\underline{\mathbf{x}})} \left[ \ln p_\theta(\underline{\mathbf{x}}|\mathbf{z}) \right] \right). \tag{9}$$

The complete encoding/decoding structure of the model depicted in Fig. 1 is trained on MSA data to minimize (9). At test time, the ELBO in (9) is used as an approximation of the log-evidence to compute the log-likelihood ratio in (1).

### 2.2 REDUCTION OF THE MODEL FOR DMS DATA

Our reduced model uses the encoding part of matVAE-MSA, and replaces the decoding part with a FC network prediction head to predict the quantitative DMS score $y \in \mathbb{R}$:

$$\mathbf{h} = \text{FC}(\text{Vec}(\text{DwFC}(\text{Transformer}(\underline{\mathbf{x}})))) \in \mathbb{R}^D,$$
$$\hat{y} = \text{FC}(\mathbf{h}) \in \mathbb{R},$$

where $\mathbf{h} \in \mathbb{R}^D$ is similar to (5). The reduced model depicted in Fig. 2 is referred to as "matENC-DMS". matENC-DMS has a capacity very close to that of matVAE-MSA since the encoder is identical in the two models, and the decoder of matVAE-MSA is symmetrical to the encoder and only aims at reconstructing the input. We argue that this is an ideal setup to test the conservation assumption often used when designing VEPs: unfit variants were selected out by nature and thus, learning a distribution over these sequences implicitly captures the biochemical constraints that characterize fit variants.

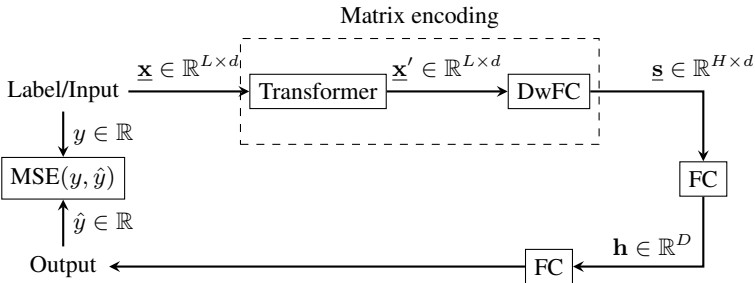

Figure 2: Model architecture of matENC-DMS. DwFC: Dimensionwise fully connected layer. CE: Cross-entropy loss. FC: Fully connected.

## 3 EXPERIMENTS

In this section we provide details on our experimental design choices. Our model comparison setup with the choice of baselines and performance metrics is explained in 3.1. The model architecture hyper-parameters are discussed in A.2, and the hyper-parameters used for training are detailed in A.3.

### 3.1 MODEL COMPARISON

**Performance metrics** The performances of our protein specific models are measured and reported on corresponding individual DMS datasets, both in terms of Spearman's Rank Correlation coefficient (SpearmanR) and area under receiver operating characteristic (AUROC). The SpearmanR measures the correlation between the ranks of the predicted scores and the ranks of the target scores. Additionally, the target score is binarized in order to compute the AUROC. To allow for a meaningful comparison of the AUROC scores, we use the binarization threshold used in ProteinGym. In brief, given a DMS dataset, a threshold on the scores is selected manually between modes in case the distribution of scores is bimodal, and as the median in case the distribution of scores is unimodal. In the rest of the paper we will primarily compare spearmanR performances.

For matENC-DMS, we train our model on DMS data with supervised learning in a 5-fold cross-validation framework. This first ensures that no variant/label pair is used for both training and testing. Secondly, this ensures that the evaluation framework is comparable to that of matVAE-MSA, with all variants in the DMS dataset used exactly once for validation. When a protein has multiple DMS datasets, both datasets were split in 5 folds and the training (resp. validation) subsets were merged. The performances on individual DMS datasets are then reported as the average of the 5 models trained on independent training sets.

**Model design choices** We experimented with mixture of diagonal Gaussian (MOG) priors with $K = 1, 10, 100$ mixture components. For $K = 1$, the mean and standard deviation of the prior distribution are fixed to $\boldsymbol{\mu}_1 = \mathbf{0} \in \mathbb{R}^D$ and $\boldsymbol{\Sigma}_1 = \text{diag}(0.01) \in \mathbb{R}^{D \times D}$. For $K = 5, 10, 100$, the trainable means are initialized randomly from a Normal distribution and for $k = 1, \ldots, K$, $\boldsymbol{\Sigma}_k = \text{diag}(0.01) \in \mathbb{R}^{D \times D}$ is fixed. We also experimented with a VAMP prior and $K = 5, 10, 100$ mixture components. The prototypes in input space were initialized randomly and all trainable. In addition to experimenting with different prior distribution hyperparameters, we performed an ablation study of our model trained on DMS data. We report the performances of models with and without Transformer(.) and DwFC(.) layers. The different models are summarized in Table 1.

| Source | Model Name | Short Description |
|---|---|---|
| Our experiments | matVAE-MSA | Matrix variational auto-encoder trained on MSA data (Fig. 1) |
| | VAMP$k$ | matVAE-MSA with a Vamp prior and $k$ components |
| | MOG$k$ | matVAE-MSA with a MOG prior and $k$ components |
| | Vec-DMS | Encoder trained on DMS data, without Transformer or DwFC |
| | DwFC-DMS | Encoder with DwFC trained on DMS data, without Transformer |
| | matENC-DMS | Matrix encoder trained on DMS data (Fig. 2) |
| ProteinGym | "Best Benchmark" | Best performing model flavor on a given DMS dataset |
| | DeepSequence | VAE-based model |
| | ESM | PLM-based model |
| | Tranception | PLM and position-wise AA frequency-based model |
| | TranceptEVE | PLM, EVE and position-wise AA frequency-based model |
| | MSA Transformer | Position-wise transformer and PLM-based model |
| | GEMME | Evolutionary tree based model |
| | VESPA | Linear Ensemble of PLM, bio-mechanic features and position-wise frequency models. |
| | MIF | PLM and inverse folding model |
| | Site-Independent | Position-wise entropy-based model |

Table 1: Summary of baselines and experiments.

**Baselines** All models with zero-shot prediction performances reported in the ProteinGym benchmark were considered for inclusion as a baseline. Only those models that demonstrated the highest SpearmanR zero-shot performances on at least one pharmacogene-related protein DMS dataset were used as a baseline. These models fall into one of the following model families: DeepSequence (Riesselman et al., 2018), ESM (Rives et al., 2021), Tranception (Notin et al., 2022a), MIF (Laine et al., 2019), GEMME (Laine et al., 2019), VESPA (Marquet et al., 2022), MSA Transformer (Rao et al., 2021). Within each family, the top-performing models vary in configuration (e.g., different parameter counts), which we refer to as model "flavors" (See Table A.6). For example ESM2 (150M), ESM2 (15B) and ESM-1v (ensemble) are all distinct flavors within the ESM family. We report the performances at the level of model families, by the average performances of the best performing model flavors of that family on individual DMS datasets. The details of which model flavor performs best on which DMS dataset are shown in Table A.5. We also compare with the "Site-Independent" model of ProteinGym, which is similar to SIFT and Polyphen-2, both still widely used in pharmacogenomics. In addition, we add a difficult baseline referred to as "Best Benchmark", which is the best performing model flavor across all model families for each DMS dataset (See Table A.6).

To compare our models trained only on DMS data, we use the supervised learning baselines from ProteinGym with 5 "Random" cross-validation splits. For this case the models belong to the following families: ESM, TranceptEVE, Tranception, DeepSequence, MSA Transformer. To the best of the authors knowledge, all these baselines consist of zero-shot prediction models trained on MSA data, and used pretrained in a supervised learning framework with embeddings of the protein sequence (See (Notin et al., 2023a)). ProteinNPT is a slightly different architecture which jointly trains on MSA and DMS data (Notin et al., 2023b). The details of the best performing model flavors are in Table A.7.

The prediction baseline models are summarized in Table 1.

## 4 RESULTS & DISCUSSION

### 4.1 CHOICE OF THE PRIOR DISTRIBUTION

Our experiments on zero-shot prediction tasks show that all our models with MOG and Vamp prior distributions have similar average SpearmanR and AUROC performances, regardless of the number of components (Fig. 3a & Table 2c). All the models we evaluated perform slightly better than MIF, Site-Independent and VESPA, but worse than all other baselines we chose (Table 2c). The standard deviation across proteins of our model is rather large, similarly to the rest of the baseline models, which prevents us from drawing strong conclusions. Numerically, our two best performing algorithms have a MOG prior with 1 and 10 components respectively, and similar reported average SpearmanR of $0.401$ and $0.400$ respectively. At the dataset level, 2/26 proteins: TPOR and SCN5A, have an increase of at least $10\%$ in SpearmanR performances for at least one latent prior with more

than one component (Table 3b). A 10% threshold for relative improvement compared to MOG1 effectively separates outliers (Fig. A.9). Overall the choice of priors did not bring the expected improvement in neither SpearmanR or AUROC performances. Numerically, the worst performing algorithm has a MOG prior with 5 components and an average SpearmanR of 0.395. For the rest of the analysis, we use the model with a MOG prior and 1 component and refer to it as "matVAE-MSA". We choose this model because it is the model with the lowest complexity and which performs best on average among our models trained on MSA data. Notably, for zero-prediction tasks, matVAE-MSA outperforms the best benchmark model for two drug target proteins: RAF1 and MK01 (Table 2b). These are two proteins involved in multiple cellular pathways, with several variants associated with the Noonan syndrome (Motta et al., 2020). Improved predictions of variant effect for these proteins can improve diagnosis and management of this syndrome, by reducing the effects of congenital heart defect and several deformities (Pandit et al., 2007).

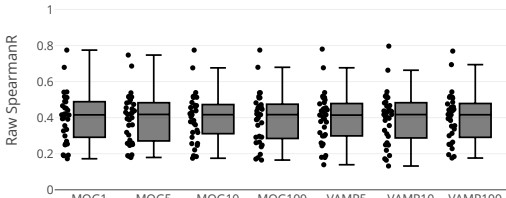

(a) Raw performances. One point denotes a DMS dataset and the boxplots describe the distribution of scores across DMS datasets.

| UnitProt ID | MOG$k$ | | | VAMP$k$ | | |
|---|---|---|---|---|---|---|
| | 5 | 10 | 100 | 5 | 10 | 100 |
| TPOR (%) | 10.3 | 32.6 | 16.8 | 22.7 | 24.4 | 1.2 |
| SCN5A (%) | 12.5 | 28.5 | -0.8 | -19.3 | -23.4 | 32.5 |

(b) Relative increase compared to MOG1. We show the 2 proteins for which the relative increase is greater than $+10\%$ for at least one choice of prior/number of components.

Figure 3: SpearmanR zero-shot performances for various choice of latent priors.

## 4.2 TRAINING ON DMS DATASETS

The protein specific models trained on DMS data (matENC-DMS) perform more than $25\%$ better than the similar model trained on MSA (matVAE-MSA) with respect to the average SpearmanR (Table A.4). Overall, all models expect Vec-DMS perform better than their zero-shot prediction counter parts (Table 2a and Table 2c). The relative increase in performances of matENC-DMS compared to matVAE-MSA across protein categories is shown in Fig. 4c. All the ADME related DMS datasets show an increase in performances with matENC-DMS. Among "Drug target" proteins, 8 (10 DMS datasets) show an increase in performances of more than $+50\%$ compared to matVAE-MSA. Since the two models have similar capacity, one explanation could be the invalidity of the conservation assumption for these specific proteins. Among other potential confounders known by the authors (e.g. quality and size of MSA or DMS, protein characteristics), none could individually explain the differences in performances according to a correlation analysis (Fig. A.10).

Our supervised matENC-DMS model outperforms, for both average SpearmanR and AUROC scores, all our chosen zero-shot prediction baselines, except "Best Benchmark", which include models with a lot more trainable parameters (Table A.4). Our ablation study shows that our model without both Transformer and DwFC layer (Vec-DMS) performs the worse overall (See also Fig. 4d). Our model with DwFC layer and without Transformer performs much better, and slightly worse than both ESM and matENC-DMS. This indicates that the transformer layer is not determinant for the good results of matENC-DMS. This could be due to the design of transformer itself, or the quality of the PDB structures predicted by AlphaFold which might put an inadequate inductive bias on the attention mechanism. Also, the models trained on DMS datasets have a larger standard deviation than the rest of the baselines for both metrics. Fig. 4b shows graphically that matENC-DMS outperforms the current zero-shot prediction "Best Benchmark" model on about half (15/33) of the DMS datasets. In terms of the protein categories, the median SpearmanR performances of matENC-DMS are below "Best Benchmark" for both "Drug target" and "ADME-other", and above for "ADME-transporter" and "ADME-CYP" (Fig. 4a). Further, the relative performances of matENC-DMS versus "Best Benchmark" varies roughly between $+75\%$ and $-75\%$ for "Drug target" (Fig. 4b). This is likely explained by the diversity of proteins included in the "Drug Target" category. The performances vary less than expected, roughly between $+25\%$ and $-25\%$, for the ADME related categories.

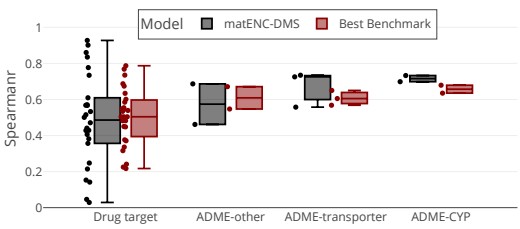

(a) Raw results for matENC-DMS (black) and "Best Benchmark" (red).

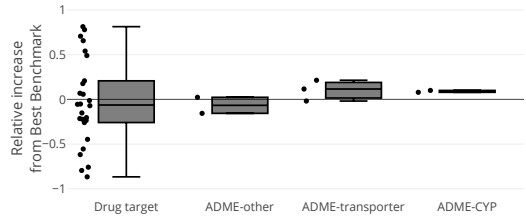
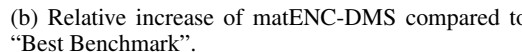

(b) Relative increase of matENC-DMS compared to "Best Benchmark".

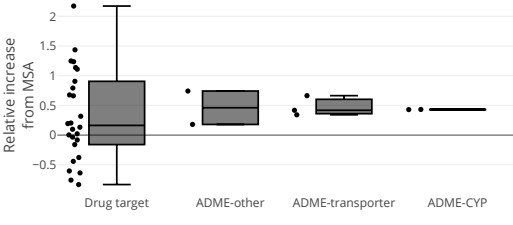

(c) Relative increase of matENC-DMS compared to matVAE-MSA.

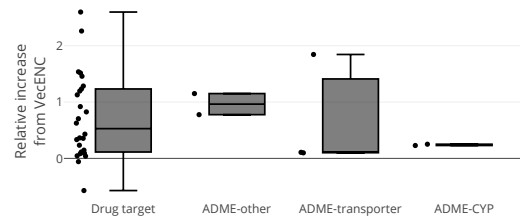

(d) Relative increase of matENC-DMS compared to Vec-MSA.

Figure 4: SpearmanR performances of matENC-DMS versus other zero-shot prediction models, per protein category. One point denotes the score obtained on a DMS dataset and the boxplots describe the distribution of scores across DMS datasets.

We show comparison with supervised learning baseline models in Table 2a. Vec-DMS performs the worse against all supervised learning baselines, while both matENC-DMS and DwFC-DMS perform better than DeepSequence on average, although the standard deviation is larger than other baselines. Overall our model perform similarly to the One-hot encoding method, but arguably worse than fine tuning approaches: TranceptEVE, MSA transformer, ESM, Tranception; and a joint training approach: ProteinNPT.

Table 2: Numerical results for our models against baselines. The models trained by us are in bold font.

| Model name | SpearmanR |
|---|---|
| Best Benchmark | $0.689 \pm 0.162$ |
| ProteinNPT | $0.671 \pm 0.195$ |
| Tranception | $0.646 \pm 0.173$ |
| ESM | $0.586 \pm 0.153$ |
| MSATransformer | $0.579 \pm 0.17$ |
| TranceptEVE | $0.536 \pm 0.156$ |
| One-Hot Encoding | $0.528 \pm 0.168$ |
| **matENC-DMS** | $0.522 \pm 0.24$ |
| **DwFC-DMS** | $0.507 \pm 0.242$ |
| DeepSequence | $0.501 \pm 0.148$ |
| **Vec-DMS** | $0.356 \pm 0.236$ |

(a) Numerical SpearmanR and AUROC ($\mu \pm \sigma$) for matENC-DMS with different architectures against baselines on supervised prediction tasks.

| UniProt ID | matVAE-MSA | Best Benchmark |
|---|---|---|
| MK01 | 0.256 | 0.241 |
| RAF1 | 0.541 | 0.482 |

(b) SpearmanR for two proteins where matVAE-MSA outperforms the Best Benchmark baseline for zero-shot prediction tasks.

| Model Name | SpearmanR | AUROC |
|---|---|---|
| Best Benchmark | $0.529 \pm 0.151$ | $0.794 \pm 0.076$ |
| ESM | $0.508 \pm 0.157$ | $0.78 \pm 0.079$ |
| TranceptEVE | $0.485 \pm 0.167$ | $0.763 \pm 0.088$ |
| Tranception | $0.478 \pm 0.165$ | $0.761 \pm 0.086$ |
| GEMME | $0.461 \pm 0.155$ | $0.750 \pm 0.085$ |
| MSA Transformer | $0.452 \pm 0.167$ | $0.746 \pm 0.089$ |
| DeepSequence | $0.425 \pm 0.151$ | $0.729 \pm 0.084$ |
| **MOG1** | $0.401 \pm 0.138$ | $0.721 \pm 0.083$ |
| **MOG10** | $0.400 \pm 0.136$ | $0.721 \pm 0.084$ |
| **VAMP100** | $0.399 \pm 0.137$ | $0.720 \pm 0.082$ |
| **VAMP10** | $0.396 \pm 0.146$ | $0.718 \pm 0.087$ |
| **MOG100** | $0.395 \pm 0.142$ | $0.717 \pm 0.085$ |
| **VAMP5** | $0.395 \pm 0.14$ | $0.716 \pm 0.083$ |
| **MOG5** | $0.395 \pm 0.137$ | $0.718 \pm 0.084$ |
| MIF | $0.394 \pm 0.185$ | $0.718 \pm 0.091$ |
| Site-Independent | $0.390 \pm 0.145$ | $0.713 \pm 0.078$ |
| VESPA | $0.383 \pm 0.147$ | $0.712 \pm 0.091$ |

(c) Numerical SpearmanR and AUROC ($\mu \pm \sigma$) for matVAE-MSA with different priors against baselines on zero-shot prediction tasks. The table is sorted vertically with respect to the SpearmanR score.

### 4.3 FUTURE WORK

Our results experimenting with expressive multi-modal priors did not show improvements compared to simple Gaussian prior. The investigations of the potential pitfalls of our current approach as well as further biology-relevant interpretations of the latent prior modes are left for future works.

Next, the joint use of both DMS and MSA data for training is an important next step towards improved model performances. In our work, our primary objective was to evaluate the information provided by MSA and DMS data separately for variant effect prediction. DMS and MSA data could nonetheless be used jointly for training, for instance in a fine tuning approach, where a VAE model is first trained on MSA data, and the encoding part is fine tuned on DMS data. This might lead to improved performances since both datasets would then contribute to the model performances. Together with experimentation on a wider range of proteins, fine tuning on DMS data is an interesting research direction that we leave as future work.

Lastly, the design of a model able to learn from multiple proteins is also an interesting next avenue for research. Our current architecture can be extended to work for proteins of different lengths $L_1, L_2, \dots$. This could be done by slightly modifying the DwFC layer (Eq. (4)). An idea would be to first define $L_M = \max(L_1, L_2, \dots)$ and initialize $\mathbf{U} \in \mathbb{R}^{H \times L_M}$. At run time, with an input protein encoding $\underline{\mathbf{x}}' \in \mathbb{R}^{L \times d}$, the first $L$ columns from matrix $\mathbf{U} \in \mathbb{R}^{H \times L_M}$ can be used, which leads to: $\underline{\mathbf{s}} = \mathbf{U}_{(:,1:L)}\underline{\mathbf{x}}' + \mathbf{b}$, where $\mathbf{U}_{(:,1:L)}$ denotes the sub-matrix of $\mathbf{U}$ which includes all the rows and the first $L$ columns of $\mathbf{U}$. The complexity of the function described in (4) depends on the length of the protein under consideration, while the memory complexity scales linearly with the length of the longest protein. This is the same as our current approach where one model is fit to individual proteins. It differs in that the weights included up to a column $l$ would be shared between all the proteins of length at least $l$. This makes the transformer learn to organize information, by placing what is relevant to all proteins in the first rows of the representation $\underline{\mathbf{x}}' \in \mathbb{R}^{H \times d}$. An issue with this approach when training on DMS data is that the meaning, support and distributions of the DMS scores vary largely across DMS datasets, thus requiring the fitness scores to be standardized (Fig. A.11). An interesting future research direction is the quantification of similarities in selection assays of DMS datasets, so that they can exploited in regression models.

## 5 CONCLUSION

We proposed a transformer-based matrix variational auto-encoder and evaluated its performances on DMS datasets of drug target and ADME proteins. We showed that advanced priors such as mixture of Gaussians and VampPrior did not provide improvement over a simple Gaussian prior for the latent space when our model was trained on MSA data (matVAE-MSA). matVAE-MSA nonetheless outperformed on average a model similar to widely used VEPs in pharmacogenomics. Moreover, matVAE-MSA outperformed the best benchmark models in ProteinGym for 2 proteins related to the Noonan syndrome. Our architecture allowed to compare performances with models of similar capacity but trained on DMS datasets (matENC-DMS) instead of MSA. Although DMS datasets were often much smaller, matENC-DMS outperformed matVAE-MSA for 15 out of 33 DMS datasets, including those of all ADME and certain drug target proteins. MSAs may thus be limiting the performances of VEPs for some proteins for which the conservation assumption does not hold. This in turn motivates the development of DMS datasets and the study of their relationships, in order to further improve variant effect prediction.

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

## A APPENDIX

### A.1 DATASETS

We train separate models on 26 pharmacogene-related proteins for which the DMS and MSA datasets are readily available from the publicly available ProteinGym repository (Notin et al., 2023a). The pharmacogene-related proteins were divided into four functional categories: Drug Targets ($n = 21$) and Absorption-Distribution-Metabolism-Excretion (ADME) related proteins ($n = 5$). The ADME category is further divided into Cytochrome ("CYP"), "transporter" and "other" ADME proteins. In total we compare performances on 33 DMS datasets, some proteins having several DMS datasets obtained under different selection assays (Table A.5).

#### A.1.1 PREPROCESSING OF MSA SEQUENCES

We followed the preprocessing steps proposed in DeepSequence for the MSA data (Riesselman et al., 2018). Sequences are removed from MSAs if they include more than $50\%$ gaps. Columns are removed from a MSA if they contain more than $30\%$ gaps across the MSA. For consistency, the columns that are removed from the MSA are also removed from the DMS datasets of that protein. When training on MSAs, each sequence is sampled with a probability proportional to the reciprocal of the number of sequences within a given Hamming distance from that sequence (Riesselman et al., 2018). A summary description of the datasets is provided in Table A.3.

Table A.3: Description of DMS and MSA datasets per protein category. The most extreme values are in bold. L: Preprocessed sequence length; MSA Num Seq (resp. DMS Num Seq): number of sequences in the MSA (resp. DMS) datasets. ADME trans.: ADME Transporter.

| Category | | L | MSA Num Seq | DMS Num Seq |
|---|---|---|---|---|
| ADME CYP (n=2) | $\mu \pm \sigma$ | $490 \pm 0$ | $260849 \pm 0$ | $6256 \pm 161$ |
| | min/max | 490 / 490 | 260849 / 260849 | 6142 / 6370 |
| ADME other (n=2) | $\mu \pm \sigma$ | $204 \pm 57$ | $86361 \pm 94716$ | $3246 \pm 569$ |
| | min/max | 164 / 245 | 19387 / 153335 | 2844 / 3648 |
| ADME tran. (n=3) | $\mu \pm \sigma$ | $579 \pm 44$ | $144978 \pm 90553$ | $10491 \pm 951$ |
| | min/max | 553 / 630 | 40416 / 197259 | 9803 / 11576 |
| Drug target (n=26) | $\mu \pm \sigma$ | $498 \pm 427$ | $62331 \pm 125352$ | $3374 \pm 3134$ |
| | min/max | **31 / 1863** | **911 / 611225** | **63 / 12464** |

### A.2 MODEL ARCHITECTURES HYPER-PARAMETERS

The encoding and decoding transformers of matVAE-MSA are designed with 3 transformer layers described in (3). The embedding dimensions of all the layers are identical and equal to $d$. The use of 3 layers allows to use information from neighbors up to order 3 according to the graph defined by the attention mask. The attention mask is a thresholded distance matrix derived from the WT protein structures predicted by Alphafold2 (Jumper et al., 2021). This means that queries are allowed to attend to keys in the attention dot product if the predicted distance in 3d between the corresponding AAs is $\leq c$. We chose $c = 7$Å which had the best performances for 4 out of 5 graph neural network-based models predicting variant effects in (Gelman et al., 2021, Table S3). The structures are readily available in the ProteinGym repository as PDB files (Notin et al., 2023a). For DwFC, we chose $H = \min(H_{min}, L)$ with $H_{min} = 200$. This means that the dimension is not reduced for proteins with small enough sequence length $L \leq H_{min}$. Following the discussion in Section 2.1.1, we experimented with $H_{min} \approx d$ on some proteins, but could not get good performances. For FCB, we used a 2-layer ReLU network with 1000 and 300 neurons, and output latent space dimension $D = 50$, this is similar to the design of DeepSequence (Riesselman et al., 2018). The networks $f_\mu$ and $f_\sigma$ are both 1-layer FC ReLU networks with $D$ neurons and output dimension $D$. For stable computation of the closed form KLD, $f_\sigma$ practically outputs $\ln \sigma^2$ using an output pointwise operation: $x \mapsto \ln(\ln(e^x + 1))$. Symmetrical design choices were used for the decoding part of matVAE-MSA.

For matENC-DMS, the $D$-dimensional latent vector in the FC bottleneck is passed into a prediction head with a 2-layer fully connected ReLU network, with 50 and 25 neurons, and an output dimension of 1. The decoding parts of matVAE-MSA are not used.

A.3 MODEL TRAINING

For matVAE-MSA, the models are trained on protein specific MSAs to minimize the negative ELBO in (9). The expected reconstruction error is approximated with a 1-sample Monte Carlo method. The reconstruction error is the cross-entropy between the true $\underline{\mathbf{x}} \in \mathbb{R}^{L \times d}$ and the reconstructed $\hat{\underline{\mathbf{x}}}$. For matENC-DMS, no variational formulation is used. The loss function is the mean squared error (MSE) between the true label $y \in \mathbb{R}$ and the reconstructed label $\hat{y}$.

For all our included proteins, the loss function for matVAE-MSA is optimized using the ADAM optimizer, with a fixed learning rate $\lambda = 8e - 5$, a batch size of 256 and $300,000$ training steps. For matENC-DMS, the loss function is optimized with a fixed learning rate $\lambda = 1e - 4$, a batch size of 512 and $100,000$ training steps. The learning rates were chosen similar to the optimal one reported for a graph neural network model in (Gelman et al., 2021, Table S3). The batch sizes were chosen to obtain the most efficient use of our hardware. In matVAE-MSA, the memory footprint is mainly due to the attention matrices in the encoder and decoder transformer. We double the batch size for matENC-DMS compared to matVAE-MSA since matENC-DMS only has an encoder transformer.

A.4 PROTEINGYM BEST PERFORMING MODELS FOR A ZERO-SHOT PREDICTION TASK, VERSUS OUR MODELS FOR BOTH ZERO-SHOT PREDICTION AND SUPERVISED LEARNING TASKS.

| Model Name | SpearmanR | AUROC |
|---|---|---|
| Best Benchmark | $0.529 \pm 0.151$ | $0.794 \pm 0.076$ |
| **matENC-DMS** | $0.522 \pm 0.24$ | $0.784 \pm 0.124$ |
| ESM | $0.508 \pm 0.157$ | $0.780 \pm 0.079$ |
| **DwFC-DMS** | $0.507 \pm 0.242$ | $0.772 \pm 0.125$ |
| TranceptEVE | $0.485 \pm 0.167$ | $0.763 \pm 0.088$ |
| Tranception | $0.478 \pm 0.165$ | $0.761 \pm 0.086$ |
| GEMME | $0.461 \pm 0.155$ | $0.750 \pm 0.085$ |
| MSA Transformer | $0.452 \pm 0.167$ | $0.746 \pm 0.089$ |
| DeepSequence | $0.425 \pm 0.151$ | $0.729 \pm 0.084$ |
| **matVAE-MSA** | $0.401 \pm 0.138$ | $0.721 \pm 0.083$ |
| MIF | $0.394 \pm 0.185$ | $0.718 \pm 0.091$ |
| Site-Independent | $0.390 \pm 0.145$ | $0.713 \pm 0.078$ |
| VESPA | $0.383 \pm 0.147$ | $0.712 \pm 0.091$ |
| **Vec-DMS** | $0.356 \pm 0.236$ | $0.695 \pm 0.123$ |

Table A.4: SpearmanR and AUROC performances ($\mu \pm \sigma$). The table is sorted vertically with respect to SpearmanR. The name of the models trained by us is in bold font. The baseline

## A.5    RAW SPEARMANR AND AUROC PERFORMANCES FOR ZERO-SHOT AND SUPERVISED LEARNING TASKS

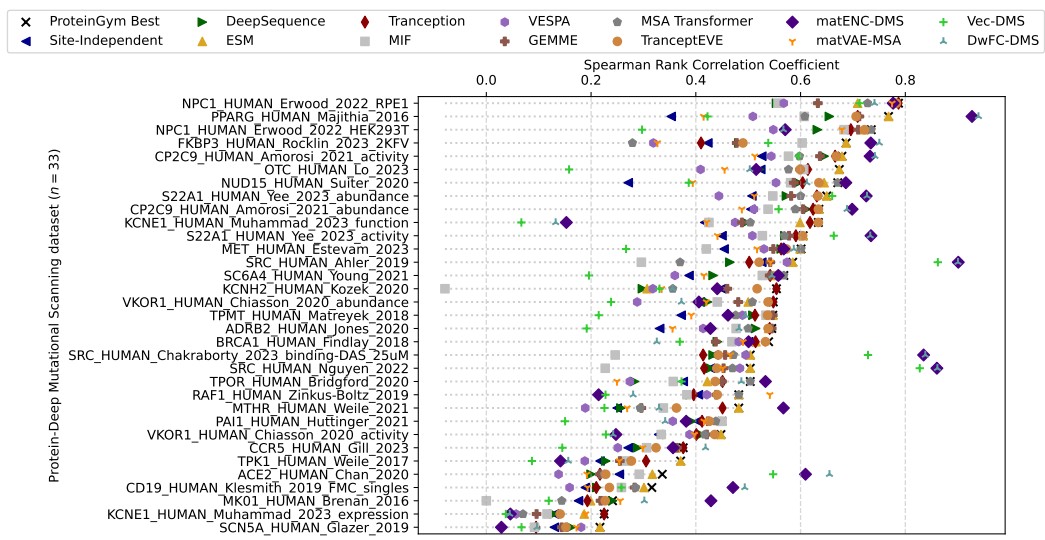

Figure A.5: Raw Spearman Rank correlation coefficient performances on all DMS datasets for zero-shot prediction task. We display the performances of our models against the best performing models in ProteinGym for pharmacogene-related proteins. The datasets are sorted in decreasing "best benchmark" performances.

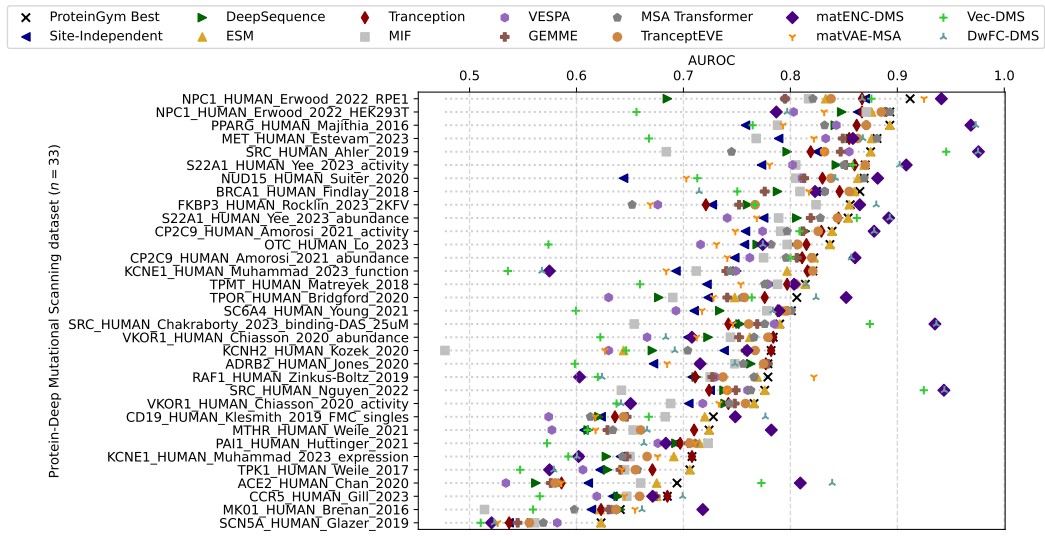

Figure A.6: AUROC performances on all DMS datasets for zero-shot prediction task. We display the performances of our models against the best performing models in ProteinGym for pharmacogene-related proteins. The datasets are sorted in decreasing "best benchmark" performances.

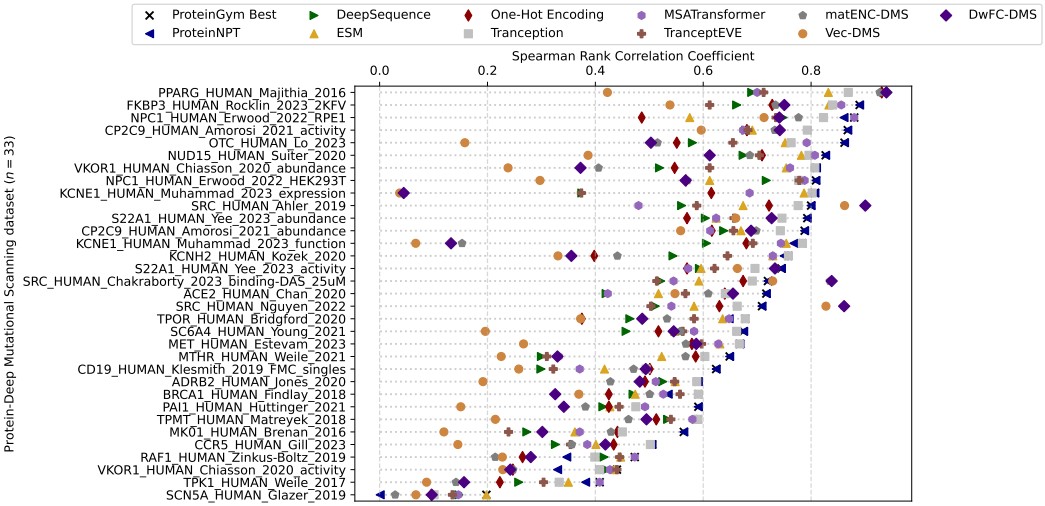

Figure A.7: SpearmanR performances on all DMS datasets for a supervised learning task. We display the performances of our models against the best performing models in ProteinGym for pharmacogene-related proteins. The datasets are sorted in decreasing "best benchmark" performances.

## A.6 PERFORMANCES WITH ASSAY SELECTION TYPE CATEGORY

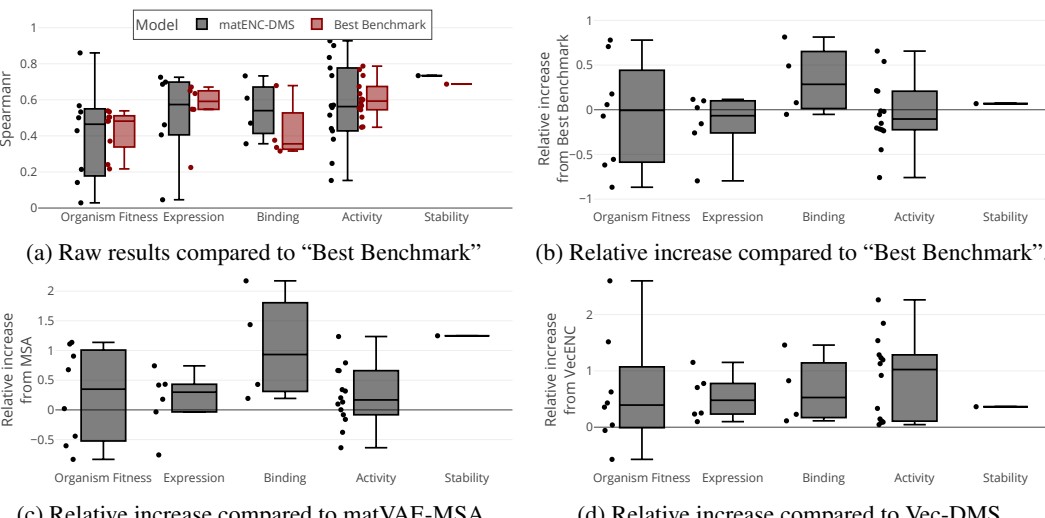

(a) Raw results compared to "Best Benchmark"

(b) Relative increase compared to "Best Benchmark".

(c) Relative increase compared to matVAE-MSA.

(d) Relative increase compared to Vec-DMS.

Figure A.8: SpearmanR performances of matENC-DMS versus other models per Selection Type sub-groups.

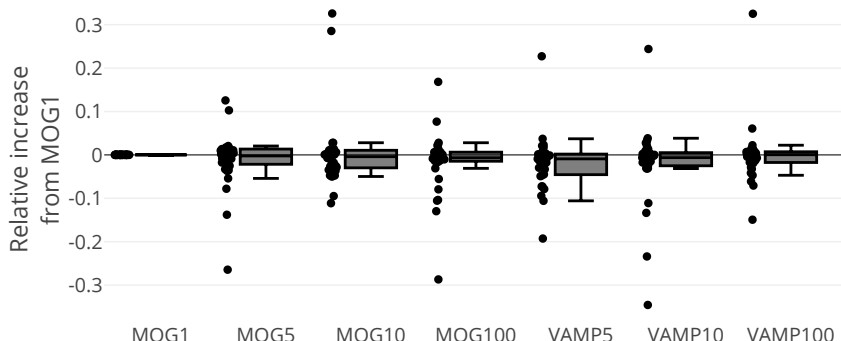

Figure A.9: Raw performances relative to MOG1. One point denotes a DMS dataset and the box-plots describe the distribution of scores across DMS datasets.

## A.7    STUDYING CONFOUNDERS FOR RELATIVE INCREASE FROM MSA

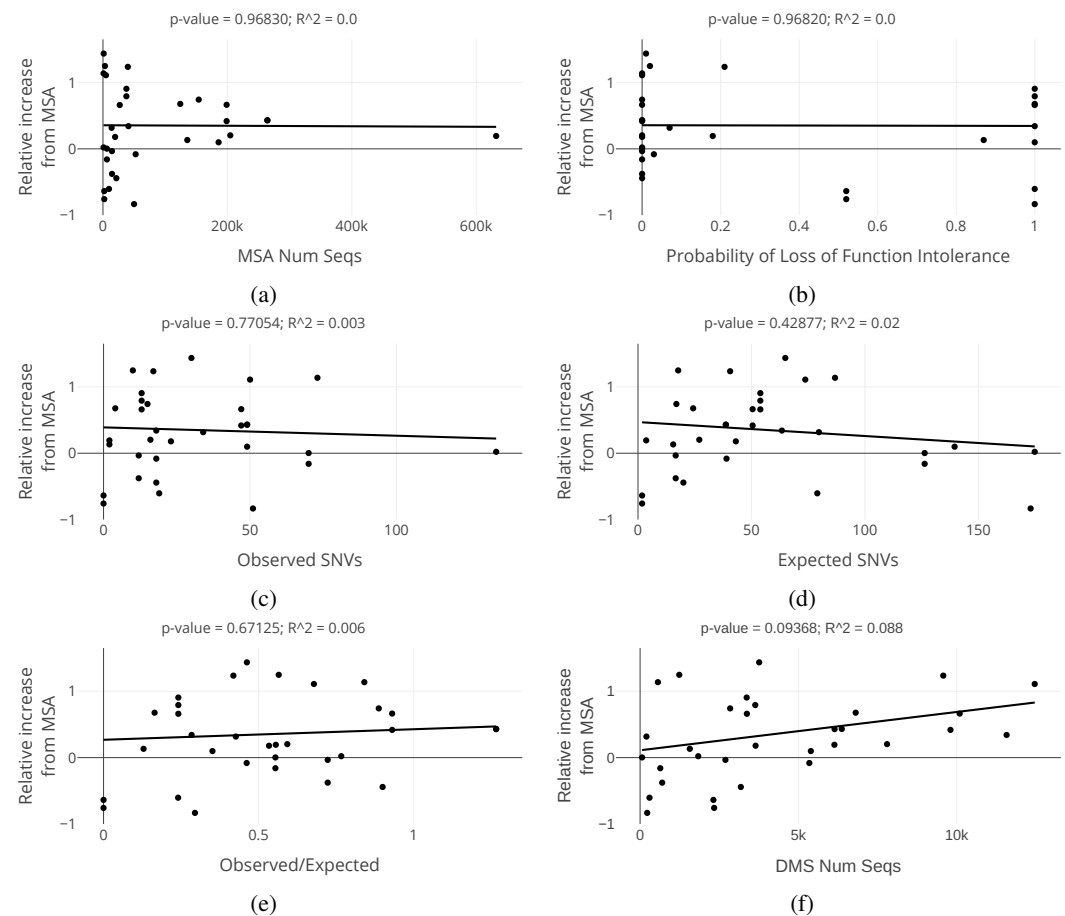

Figure A.10: Univariate correlation analysis for potential confounders of the relative increase of matENC-DMS from matVAE-MSA. None of the considered confounders are significant to explain the relative increase from MSA. A multivariate correlation analysis was performed and did not show any significance (not shown). **MSA Num Seqs**: Number of variants in MSA; **Probability of Loss of Function Intolerance**: Genes with a pLI close to 1 are often associated with haploinsufficiency and dominant genetic diseases; **Expected (resp. Observed) SNVs**: Expected (resp. Observed) Single-nucleotide variant in each gene; **Observed/Expected (o/e)**: Constrained genes have fewer observed variants than expected (low o/e) and are under a higher degree of selection than less constrained genes. **DMS Num Seqs**: Number of variants in DMS data;

## A.8 ADDITIONAL DATASET INFORMATION

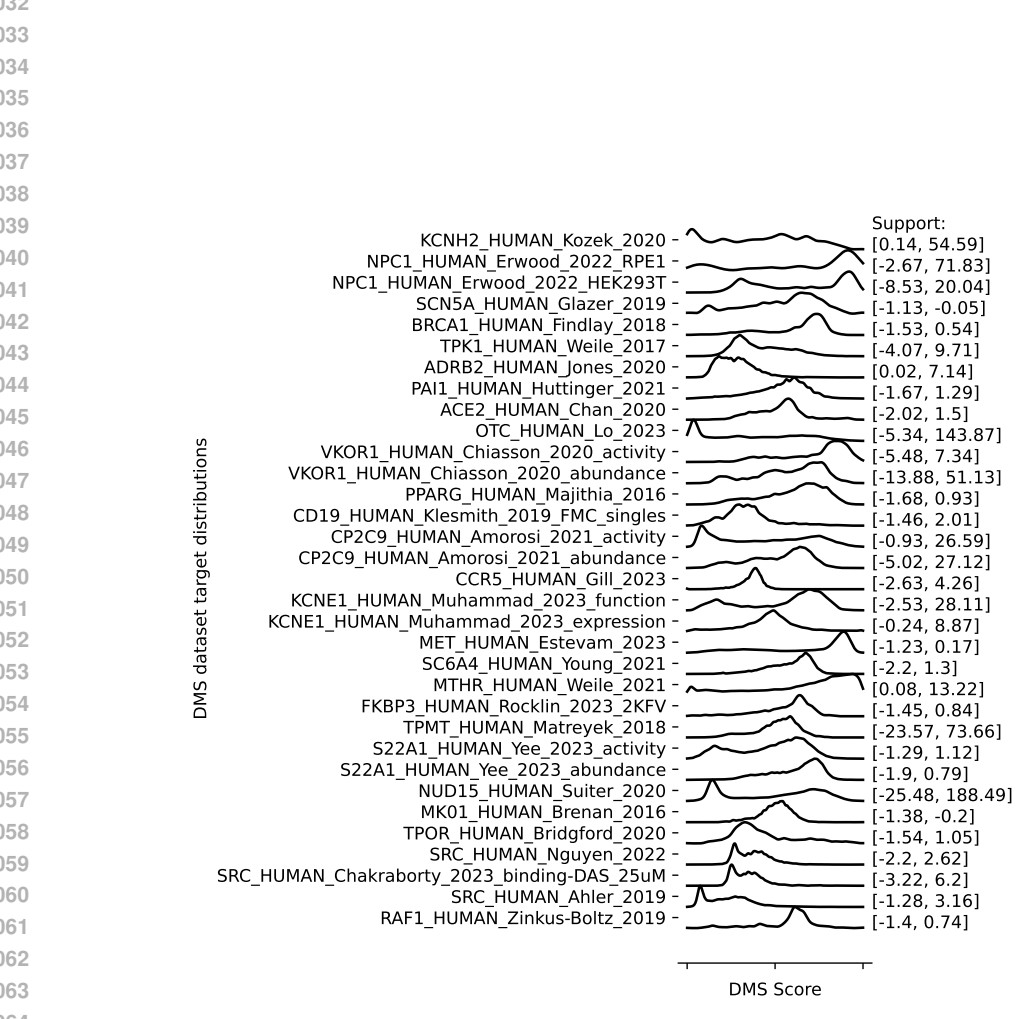

Figure A.11: Empirical kernel density estimates of the density function of DMS datasets target score. The support and range of each density function are standardized to [0,1] for visualization. The true support of the density functions is annotated on the right hand side. The densities are displayed with an increment of 1 along the y-axis.

| Category | UniProt ID | L | MSA Num Seq | DMS Num Seq | Selection Type | Best Benchmark - Flavor |
|---|---|---|---|---|---|---|
| ADME CYP | CP2C9 | 490 | 260849 | 6142 | Binding | ESM2 (150M) |
| | | | | 6370 | Expression | ESM2 (650M) |
| ADME other | NUD15 | 164 | 153335 | 2844 | Expression | MSA Transformer (ensemble) |
| | TPMT | 245 | 19387 | 3648 | Expression | ESM-1v (ensemble) |
| ADME transporter | S22A1 | 553 | 197259 | 9803 | Expression | ESM-1v (ensemble) |
| | | | | 10094 | Activity | ESM-1v (ensemble) |
| | SC6A4 | 630 | 40416 | 11576 | Activity | MSA Transformer (ensemble) |
| Drug target | ACE2 | 805 | 10865 | 2223 | Binding | MIF |
| | ADRB2 | 413 | 201108 | 7800 | Activity | GEMME |
| | BRCA1 | 1863 | 974 | 1837 | Organismal Fitness | VESPA |
| | CCR5 | 352 | 611225 | 6137 | Binding | Tranception L |
| | CD19 | 556 | 1171 | 3761 | Binding | MIF |
| | FKBP3 | 69 | 3211 | 1237 | Stability | ESM-IF1 |
| | KCNE1 | 129 | 2104 | 2315 | Activity | TranceptEVE L |
| | | | | 2339 | Expression | Tranception M no retrieval |
| | KCNH2 | 31 | 13900 | 200 | Activity | Tranception M |
| | MET | 287 | 184827 | 5393 | Activity | MSA Transformer (ensemble) |
| | MK01 | 360 | 123422 | 6809 | Organismal Fitness | DeepSequence (ensemble) |
| | MTHR | 656 | 4724 | 12464 | Organismal Fitness | ESM2 (150M) |
| | NPC1 | 1278 | 6234 | 63 | Activity | Tranception S |
| | | | | 637 | Activity | MSA Transformer (ensemble) |
| | OTC | 354 | 134484 | 1570 | Activity | ESM-IF1 |
| | PAI1 | 402 | 51792 | 5345 | Activity | MIF-ST |
| | PPARG | 505 | 39639 | 9576 | Activity | ESM2 (15B) |
| | RAF1 | 648 | 9609 | 297 | Organismal Fitness | MSA Transformer (single) |
| | SCN5A | 32 | 49959 | 224 | Organismal Fitness | ESM-1v (single) |
| | SRC | 536 | 37311 | 3366 | Organismal Fitness | ESM-1v (ensemble) |
| | | | | 3372 | Activity | ESM-1v (ensemble) |
| | | | | 3637 | Activity | ESM-1v (ensemble) |
| | TPK1 | 243 | 21338 | 3181 | Organismal Fitness | ESM2 (15B) |
| | TPOR | 635 | 911 | 562 | Organismal Fitness | MSA Transformer (single) |
| | VKOR1 | 163 | 14425 | 697 | Activity | ESM-1v (ensemble) |
| | | | | 2695 | Expression | Tranception L |

Table A.5: Deep Mutation Scanning datasets details retrieved from ProteinGym. **Uniprot ID**: Universal protein identifier; **L**: Preprocessed sequence length; **MSA Num Seq** (resp. **DMS Num Seq**): number of sequences in the MSA (resp. DMS) datasets. **Selection Type**: DMS assay selection type. **Best Benchmark - Flavor**: best performing model flavor for zero-shot prediction tasks.

| Best Benchmark - Family | Best Benchmark - Flavor |
|---|---|
| ESM (n=15) | ESM2 (15B) x2 |
| | ESM-1v (ensemble) x7 |
| | ESM2 (150M) x2 |
| | ESM2 (650M) x1 |
| | ESM-IF1 x2 |
| | ESM-1v (single) x1 |
| MSA Transformer (n=6) | MSA Transformer (single) x2 |
| | MSA Transformer (ensemble) x4 |
| Tranception (n=5) | Tranception L x2 |
| | Tranception S x1 |
| | Tranception M x1 |
| | Tranception M no retrieval x1 |
| MIF (n=3) | MIF-ST x1 |
| | MIF x2 |
| DeepSequence (n=1) | DeepSequence (ensemble) x1 |
| TranceptEVE (n=1) | TranceptEVE L x1 |
| GEMME (n=1) | GEMME x1 |
| VESPA (n=1) | VESPA x1 |

Table A.6: Summary of best model flavors and families for zero-shot prediction tasks.

| Best Benchmark - Family | Best Benchmark - Flavor |
|---|---|
| ProteinNPT (n=23) | ProteinNPT x23 |
| Tranception (n=5) | Tranception Embeddings x5 |
| MSA Transformer (n=2) | MSA Transformer Embeddings x2 |
| ESM (n=1) | ESM-1v + One-Hot Encodings x1 |
| MSA Transformer + One-Hot Encodings (n=1) | MSA Transformer + One-Hot Encodings x1 |
| TranceptEVE (n=1) | TranceptEVE + One-Hot Encodings x1 |

Table A.7: Summary of best model flavors and families for supervised prediction tasks.

