# OpenReview forum: "A Matrix Variational Auto-Encoder for Variant Effect Prediction in Pharmacogenes"
_ICLR.cc/2025/Conference — Submitted to ICLR 2025_

### Official Review · Reviewer_QCTd · 2024-10-31

**Soundness:** 3
**Presentation:** 2
**Contribution:** 1
**Rating:** 3
**Confidence:** 4

**Summary:**

This paper first proposes an unsupervised variant effect predictor (VEP), matVAE-MSA. The model is a VAE with transformer layers for the encoder and decoder with a sparse attention mask derived from the WT AlphaFold-predicted structure. Additionally the authors propose a supervised model, matENC-DMS which uses the same encoding architecture as matVAE-MSA, but maps the encoded vector to a scalar to predict the fitness label. matVAE-MSA does not consistently outperform existing unsupervised approaches, but does provide the best results for two proteins, MK01 and RAF1. matENC-DMS, which is trained on family-specific DMS data, consistently improves performance over unsupervised approaches.

**Strengths:**

The experimental results are cleanly laid out and the authors do a nice job of not over-selling their results.

**Weaknesses:**

Unfortunately I see a number of weaknesses with the current paper. First, the use of a transformer as a VAE for modeling protein families is not new and exists in ProT-VAE. Nonetheless, given that this new architecture does not outperform existing simple VAE architectures it is not clear what contribution this paper is making with this? While the matENC-DMS results are interesting, there is no benchmarking with other supervised approaches. In particular, the paper "Learning protein fitness models from evolutionary and assay-labeled data" addresses combining evolutionary data with DMS data. At a minimum the authors should compare to this approach. Overall, I wasn't able to see what contributions the authors are making. The new VAE model doesn't seem to add any new insights into how to design better unsupervised generative models and it does not outperform existing approaches. The supervised model is not benchmarked against any similar approaches and does not provide any new insights.

**Questions:**

Can the authors more clearly explain what they feel are the contributions of the paper?

---

> ### Author Response · Authors · 2024-11-20
> **Authors response**
>
> We would like to thank you for the thorough review and the interesting points raised. For simplicity of reading, we quote the review and provide our answers in bullet points.
>
>
> *Unfortunately I see a number of weaknesses with the current paper. First, the use of a transformer as a VAE for modeling protein families is not new and exists in ProT-VAE.*\
>     • We thank the reviewer for pointing this out, we were not aware of Prot-VAE (submitted as a preprint in January 2023: https://www.biorxiv.org/content/10.1101/2023.01.23.525232v1). We find the design of the latent space however not entirely clear from the preprint, therefore we refrain from further comparison at this stage.
>
> *Nonetheless, given that this new architecture does not outperform existing simple VAE architectures it is not clear what contribution this paper is making with this?*\
>     • Thank you for this comment. We agree that the contributions for our MSA based model should have been stated more clearly. Although our architectures do not outperform most baselines models on average, we found that our zero-shot prediction model outperforms other methods for 2 proteins, both related to the Noonan syndrome. We now state this more clearly in the results section as well as in the abstract.
>
> *While the matENC-DMS results are interesting, there is no benchmarking with other supervised approaches. In particular, the paper "Learning protein fitness models from evolutionary and assay-labeled data" addresses combining evolutionary data with DMS data. At a minimum the authors should compare to this approach. Overall, I wasn't able to see what contributions the authors are making. The new VAE model doesn't seem to add any new insights into how to design better unsupervised generative models and it does not outperform existing approaches. The supervised model is not benchmarked against any similar approaches and does not provide any new insights.*\
>     • Thank you for this observation. We now provide a comparison with supervised learning approach reported in ProteinGym in Table 2.a.
> These methods are fine tuning of models trained on MSA so the comparison with our methods is not exactly fair: but we agree that the comparison is an interesting addition to the paper.
>
> *Questions:
> Can the authors more clearly explain what they feel are the contributions of the paper?*\
>     • We would like to summarize our contributions as follows. The first contribution of our work is a set of experiments on VAEs with novel input layer (based on transformers and a linear layer with shared weights inspired from linear matrix decomposition). These experiments show that we can improve the current baseline for 2 proteins related to the Noonan syndrome (We added this observation to the revised paper). Next we perform experiments with expressive MOG and VAMP priors and show that only 2 proteins benefit from it. Lastly we show that our supervised learning model performs better than its MSA counter parts for all ADME proteins and some drug targets. This shows that some drug targets are equally well predicted by zero-shot prediction models compared to supervised models. This can be due to the amount of DMS data available or other protein related characteristics.

---

> > ### Comment · Reviewer_QCTd · 2024-11-20
> > **Response to authors**
> >
> > Thank you for the reply and for the new experiments.
> >
> > Showing an improvement on 2 / 33 proteins is not sufficient enough of a contribution. If I take any of the top-performing models and modify a hyper parameter like the size of a hidden layer, with very high probability I will see an improvement on at least 2 / 33 proteins. This is why the current standard involves some sort of average or median spearman across all datasets. Another way to phrase the results is that compared to current VAE approaches, this method performs worse on ~31/33 proteins.
> >
> > Regarding the supervised learning experiments, with the new baseline it is clear that there are no improvements over existing approaches. I disagree that fine-tuning of models trained on MSA is not exactly fair as this is a reasonable approach to tackling this problem since DMS data typically only exists for single-mutations so some amount of prior information is needed to predict fitness at unobserved positions. Furthermore, you too can fine-tune your MSA-pretrained model too.
> >
> > I will keep my score.

---

### Official Review · Reviewer_t6KY · 2024-11-02

**Soundness:** 2
**Presentation:** 1
**Contribution:** 1
**Rating:** 3
**Confidence:** 4

**Summary:**

This paper proposes an encoder-decoder language model similar to DeepSequence for the task of mutation effect prediction and tests it on 33 drug-related DMS datasets from ProteinGym.

**Strengths:**

- A clear presentation on the new transformer-based module for the encoder and decoder.
- A comprehensive investigation and analysis on the impact of different designed modules to the prediction task.

**Weaknesses:**

- The presentation of the motivation is unclear (Q1, 3).
- The justification for the experimental design and significance of the results is not clearly articulated (Q2, 3, 4, 6).
- The design of the prediction tasks appears to be questionable (Q3, 5, 8).
- The comparison with baseline methods is incomplete (Q7).

**Questions:**

1. If pharmacogenes experience low evolutionary pressure, why is MSA included? (line 18)
2. What is the significance of the statement "sets a new benchmark for 2 out of 26 proteins"? (lines 21-22)
3. Regarding the "5-fold cross-validation framework," what is the rationale for training a model to fit a DMS dataset rather than conducting zero-shot predictions? For a new protein, if a DMS dataset is already available, it is likely that users have already identified favorable mutants.
4. Are there any differences between the MSA sequences used and those provided by the ProteinGym benchmarks? If so, why not utilize ProteinGym's versions?
5. Why is there a manual selection of a threshold to binarize the DMS target scores? What are the specific criteria for this selection and what are the resulting values?
6. Could you clarify the statement "For instance ESM2 (150M), ESM2 (15B) and ESM-1v (ensemble) are all flavors of ESM"?
7. How should we interpret the statement "We compared with those models in ProteinGym which perform the best on at least one pharmacogene-related protein DMS dataset according to SpearmanR"? Does "perform the best" refer to zero-shot prediction or supervised learning tasks? What are the candidate models and their respective scores? Have all models on the ProteinGym leaderboard been considered in this comparison?
8. While the algorithm does not apply any specific designs to pharmacogene, it seems like a general framework that can be applied to any mutation effect prediction tasks. In this case, it is suggested to test the model on the complete ProteinGym benchmarks of all 217 assays.

---

> ### Author Response · Authors · 2024-11-15
> **Clarification of question 3**
>
> Dear reviewer,
>
> thank you very much for your valuable feedback and comments on our first submission.
> We are working on a detailed global response to your reviews, and on an updated, better manuscript.
>
> Towards this goal and in order to take your observations into account in the best possible way, I would like to ask you to provide a clarification of the observation after the question mark in question 3: "For a new protein, if a DMS dataset is already available, it is likely that users have already identified favorable mutants.". Thank you!

---

> > ### Comment · Reviewer_t6KY · 2024-11-18
> > **Clarification on Q3**
> >
> > Building a DMS database typically encompasses at least thousands of mutants on a protein. While the positive rate is considerably low (owing to the randomness of mutations), there invariably exists a portion of positive mutants, which I refer to as *favorable mutants* in this context. Here I call them favorable mutants to indicate that they are positive, rather than that they are sufficiently optimized for practical applications (after all you usually need to take multiple iterations of experiments to get a sufficiently good mutant). The key point here is that **the majority of proteins lack a DMS database to train a supervised learning model**. They might only have a handful, or even no publicly available mutants. This is also the reason why the general approach with ProteinGym leans towards zero-shot prediction and employs Spearman's correlation as the evaluation metric, instead of building a supervised learning task.
> >
> > I hope the explanation makes my question clear now.

---

> ### Author Response · Authors · 2024-11-20
> **Authors response 1/2**
>
> We would like to thank the reviewer for the relevant points raised
> Questions:
>
> *1)*\
>        ◦ Thank you for this question. We included MSA because all state of the art VEPs in the literature are trained with MSA, and we wish to perform comparison between MSA and DMS data.
>
> *2)*\
>         ◦ Thank you for this question. We added some biological interpretation of this finding in the results section 4.1: “RAF1 and MK01 are two proteins involved in multiple pathways. Several variants of RAF1 and MK01 are associated with the Noonan syndrome, causing congenital heart defect and several deformities (Motta et al., 2020; Pandit et al., 2007). Improved predictions of variant effect for these proteins can improve diagnosis and management of this syndrome.”
>
> *3)*\
>  ◦ Thank you for this question (and for the clarification). Our motivation for training on DMS datasets is that it provides a way to test the evolutionary assumption: correlation between function (defined by the selection assay used for DMS) and “fitness” (defined by the fact that a variant is present or not in a MSA). Because the model we use for training on DMS data has similar capacity than the model we use for training on MSA (the former is the encoding part of the latter), our experiment provides some insight on the relevant of training on MSA for pharmacogenes.  We agree with your comment that the scarce DMS datasets is an issue in general which might limit and bias performances of purely supervised methods.
>
> *4)*\
> ◦ Thank you for this question. We do use proteingym’s version and this is mentioned in the original submission in the “datasets” section (now moved to the appendix): “We train separate models on $26$ pharmacogene-related proteins for which the DMS and MSA datasets are readily available from the publicly available ProteinGym repository  (Notin et al., 2023).”. We also use similar preprocessing steps introduced in the development of DeepSequence (details are now in section A.1.1).
>
> *5)*\
> ◦ Thank you for this question. I have clarified the binarization method in the new submission: “Additionally, the target score is binarized in order to compute the AUROC. To allow for a meaningful comparison of the AUROC scores, we use the binarization threshold used in ProteinGym (Notin et al., 2023). In brief, a threshold on the target score is selected manually between modes in case the distribution of scores is bimodal, and as the median in case the distribution of scores is unimodal.  In the rest of the paper we will primarily compare spearmanR performances.”. We do not provide the exact values in the paper but they can be found in the columns “DMS_binarization_” of the protein gym reference file: https://github.com/OATML-Markslab/ProteinGym/reference_files/DMS_substitutions.csv

---

> > ### Author Response · Authors · 2024-11-20
> > **Authors response 2/2**
> >
> > *6)*\
> > ◦ Thank you for this comment, we clarified the statement describing the baseline model families and what we call flavors. “All models with performances reported in the ProteinGym benchmark were considered for inclusion as a baseline. Only those models that demonstrated the highest SpearmanR zero-shot  performances on at least one pharmacogene-related protein DMS dataset were considered as a baseline. These models fall into one of the following model families: DeepSequence (Riesselman et al., 2018), ESM (Rives et al., 2021), Tranception (Notin et al., 2022a), MIF (Laine et al., 2019), GEMME (Laine et al., 2019), VESPA (Marquet et al., 2022), MSA Transformer (Rao et al., 2021). Within each family, the top-performing models vary in configuration (e.g., different parameter counts), which we refer to as model ”flavors” (See Table A.4). For example ESM2 (150M), ESM2 (15B) and ESM-1v (ensemble) are all distinct flavors within the ESM family. We report the performances at the level of model families, by the average performances of the best performing model flavors of that family on individual DMS datasets.”
> >
> > *7)*\
> > ▪ Thank you for this question. This statement referred to zero-shot prediction tasks only. However, we have extended our results and are now showing results on supervised learning algorithms from ProteinGym Table 2.a. The model names are mentioned in Table A.5 and Table A.6, their respective scores are shown in relation to other models in Figure A.5, and Figure A.6, and in Figure A.7 for the supervised learning models. Additionally, all the models we compare with are described in the Related works section.
> > All models have been considered and the ones not performing the best at least once on our proteins were included. We argue that comparing with those models performing best on at least one pharmacogene is more relevant than comparing only with models in the overall leaderboard. This is because those models might have the highest performances on other families of proteins that are not under study here, meaning that we might miss out relevant comparisons with the highest performing model on pharmacogenes.
> >
> > *8)*\
> > We thank the reviewer for this comment. We do acknowledge that there are no conceptual difficulties in testing the model on all 217 assays, this is planned as future work and we now mention this in the future works section. However, our paper focuses on pharmacogene-related proteins and we thus considered other proteins and assays out of scope.

---

> > > ### Comment · Reviewer_t6KY · 2024-11-29
> > >
> > > Thank you for your response. However, most of my concerns, especially the lack of testing on ProteinGym, have not been fully addressed. After reading the opinions of other reviewers, I am inclined to maintain my original score.

---

### Official Review · Reviewer_3ZxL · 2024-11-03

**Soundness:** 1
**Presentation:** 1
**Contribution:** 1
**Rating:** 3
**Confidence:** 3

**Summary:**

Here the authors propose a framework based on representation learning via VAEs for use on downstream deep mutational scanning tasks. The authors discuss their design choices and then proceed to benchmark their method against standard baseline methods for these tasks.

**Strengths:**

* **Novelty**: To the best of my knowledge, the authors' proposed framework is novel.
* **Impact**: The authors' motivation (i.e., assessing how well evolutionary pressure corresponds to fitness and the corresponding impact on variant effect prediction) is solid, and such studies would likely be of interest to the machine learning for proteins community.

**Weaknesses:**

Despite the potential impact of the authors' work, I believe that the authors' submission has significant issues that prevent me from recommending acceptance at this time. I provide details on the major issues below:

* **Unclear motivation for model design choices**: The authors spend a significant amount of time experimenting certain modeling/architecture choices (e.g. using a mixture of gaussians prior rather than a unimodal prior), which in the end don't have an impact on model performance. Indeed, this is listed as one of the authors' main contributions in the introduction. Could the authors comment on their rationale for exploring these modeling choices? Did previous results for this task find that more expressive priors led to improved performance? Or was there a more principled reason to assume that these specific priors would lead to better performance? Without more context it's hard for the reader to understand why these results are being presented. On a related note, it would be great to see an ablation study assessing the impact of training a predictor on DMS datasets using representations from previous methods (e.g. DeepSequence) compared to the same task with the authors' proposed architecture. Without this information, it's difficult for the reader to understand whether any boosts in performance for the models trained on DMS data can be attributed to the authors' proposed encoder network or if the results are solely due to training on DMS data.
* **Unclear significance of experimental results**: Perhaps most importantly, it's not clear to me that any meaningful conclusions can be drawn from the experimental results (e.g. those presented in Table 3). In particular, given the large error bars it's difficult for the reader to assess if the provided results are statistically significant. Could the authors provide results from e.g. a t test? Moreover, it's not clear to me how the authors selected their final model hyperparameters (e.g. learning rates). The authors mentioned that their choices "preserv[ed] stability and convergence", but without more details it's hard to tell if these values were cherry-picked. Were these parameters e.g. chosen via cross-validation/performance on a held-out validation set? Given these issues, it's thus unclear whether the authors' claims are supported by their experimental results.
* **Not self-contained/writing issues**: Given that ICLR is a general machine learning conference (as opposed to a more biology-focused venue), it would greatly improve the manuscript for the authors to spend more time in the introduction describing the problem setup and significance. Indeed, the introduction section to the manuscript feels extremely rushed, with little time spent on introducing the problem setting tackled by the authors. For example, providing a gentler introduction to domain-specific terms like deep mutational scanning, On the other hand, a significant amount of space is spent describing hyperparameters (e.g. sections 3.1/3.2) or details of individual datasets (e.g. section 2.3 + Table 1), which could be relegated to the Appendix. I would thus recommend that the authors restructure the manuscript so that the main text is self-contained for a general ICLR reader, with ancillary experimental details moved to the appendix to make space as needed. On a related note that could save some space, it's unclear to me why the authors spend a significant amount of time introducing certain mathematical/machine learning concepts which are subsequently not used in the method (e.g. introducing matrix decomposition before stating that transformer layers are used).

**Questions:**

See "Weaknesses"

---

> ### Author Response · Authors · 2024-11-20
> **Authors response 1/2**
>
> We would like to thank the reviewer for clear, concise and relevant comments. Please find our answers below
>
> *Could the authors comment on their rationale for exploring these modeling choices? Did previous results for this task find that more expressive priors led to improved performance? Or was there a more principled reason to assume that these specific priors would lead to better performance? Without more context it's hard for the reader to understand why these results are being presented.*\
> ◦ Thank you for this comment. We do acknowledge that this should have been introduced earlier and with clearer references. The use of MOG was motivated in the methods (section 2.1.2): “As opposed to Gaussian distribution, MOG are multimodal and add more structure in latent space. This in turn can lead to a more expressive generative model, with a latent space able to important differences in input space in different modes, and fully use individual modes to encode subtle differences.” In addition to being in principle more expressive, it was shown that multimodal priors improved clustering tasks on image data. We now added a paragraph in our reworked introduction.
>
> *On a related note, it would be great to see an ablation study assessing the impact of training a predictor on DMS datasets using representations from previous methods (e.g. DeepSequence) compared to the same task with the authors' proposed architecture. Without this information, it's difficult for the reader to understand whether any boosts in performance for the models trained on DMS data can be attributed to the authors' proposed encoder network or if the results are solely due to training on DMS data.*\
> ◦ Thank you for this suggestion. Exploring the potential performance boost of finetuning is an interesting avenue. However this paper focuses on the impact of MSA vs DMS training and therefore we argue that it is more relevant to train from scratch on these two datasets with a similar model architecture. We however acknowledge that our comparison with only zero-shot prediction methods is inadequate, since the models we compare with have not been trained at all on DMS data. We now display our results against supervised learning baselines in Table 2.a.
>
> *Unclear significance of experimental results: Perhaps most importantly, it's not clear to me that any meaningful conclusions can be drawn from the experimental results (e.g. those presented in Table 3). In particular, given the large error bars it's difficult for the reader to assess if the provided results are statistically significant. Could the authors provide results from e.g. a t test?*\
> ◦ We thank the reviewer for this suggestion. We have not seen t-tests performed in the literature when comparing such models, moreover the Gaussian assumption for the samples of t-tests is likely too strong. To clarify our claims, we now explicitly mention the large standard deviation across DMS datasets (Section 4.1 in revised manuscript). At the protein level, we claim that only two proteins benefit from more expressive prior on the basis that their performances increases more than 10%. We chose 10% because it is a good threshold on the relative increase to separate outliers, this is shown on a new figure: Figure A.9
>
> *Moreover, it's not clear to me how the authors selected their final model hyperparameters (e.g. learning rates). The authors mentioned that their choices "preserv[ed] stability and convergence", but without more details it's hard to tell if these values were cherry-picked. Were these parameters e.g. chosen via cross-validation/performance on a held-out validation set? Given these issues, it's thus unclear whether the authors' claims are supported by their experimental results.*\
>     • We thank the reviewer for pointing this out. Our hyperparameters were chosen similarly to what we mention regarding the cutoff threshold for the amino acid distance matrix in Section “Model Architectures hyper-parameters”, i.e. on the basis of the study performed by Gitter et al.. In the paragraph related to the learning rate (now moved to appendix), we mentioned that we confirmed experimentally that the learning rates we chose led to convergence and stable training. The paragraph has been rephrased to avoid confusion: “The learning rates were chosen similar to the optimal one reported for a graph neural network model in \citep[Table~S3]{gelmanNeuralNetworksLearn2021}. The batch size was chosen to obtain the most efficient use of our hardware memory capacity. In matVAE-MSA, the memory footprint is mainly due to the attention matrices in the encoder and decoder transformer. We double the batch size for matENC-DMS compared to matVAE-MSA since matENC-DMS only has an encoder transformer.”

---

> > ### Author Response · Authors · 2024-11-20
> > **Authors response 2/2**
> >
> > *Not self-contained/writing issues:*\
> > ◦ We thank the reviewer for these suggestions. We do agree that not enough space was allocated to motivating design choices, and that some more background on the task at head are warranted. We have now reworked our introduction with slightly more biology background, a new paragraph motivating the use of more expressive priors, and a new paragraph discussing recent work (from 2024) which proposes joint training of MSA and DMS data.
> >
> > *On a related note that could save some space, it's unclear to me why the authors spend a significant amount of time introducing certain mathematical/machine learning concepts which are subsequently not used in the method (e.g. introducing matrix decomposition before stating that transformer layers are used).*\
> > ◦ Thank you for this comment. The transformer is part of the matrix encoding block which is inspired from linear matrix decomposition. We feel that it is more appropriate to discuss the linear decomposition first, in order to properly motivate the use of a transformer for learning relevant encoding of the data. This is done in the original submission in the last sentence before the formulation of the matrix encoding layer: “To ensure that the model is flexible enough to learn a useful encoding, we propose to learn a representation of $\underline{\mathbb{x}}$ with a transformer, prior to reducing the first dimension to a fixed $H < L$ with a trainable linear transform. ”

---

> > > ### Comment · Reviewer_3ZxL · 2024-11-21
> > >
> > > Dear authors,
> > >
> > > Thank you for your response to my concerns. Similar to Reviewer QCTd's concerns, given the apparent lack of meaningful improvements over previously proposed methods, I believe that the results in the manuscript do not represent a sufficient contribution for publication at ICLR. I am thus choosing to keep my score.

---

### Official Review · Reviewer_y9E7 · 2024-11-09

**Soundness:** 3
**Presentation:** 4
**Contribution:** 2
**Rating:** 3
**Confidence:** 4

**Summary:**

This paper introduces two new methods for variant effect prediction: (1) matVAE-MSA which is a VAE trained on sequences from the protein family of interest and (2) matENC-DMS which is trained on DMS data from a specific protein. matVAE-MSA is similar to DeepSequence and EVE but the authors introduce architectural changes (i.e. self-attention layers) and experiment with more complex priors: mixture of diagonal Gaussians and VAMP. On pharmacogenes in ProteinGym, matVAE-ESM underperforms DeepSequence and ESM models. However, matENC-DMS, whose architecture is exactly the encoder portion of the DMS data and is trained on functional activity scores from a DMS assay, outperforms all other methods. Given that models trained on DMS data do significantly better than models solely trained on sequences from a MSA, the authors conclude that DMS data is valuable to train variant effect predictors on pharmacogenes.

**Strengths:**

- The authors explore novel architectural innovations to the DeepSequence/EVE family of models. In particular, they use self-attention layers where the attention map is determined from predicted contacts in AF2 structures. They also place more expressive priors on the latent space in matVAE-MSA. These are innovative ideas that have not yet been considered in the field.

- The authors clearly benchmark their method to other state-of-the-art methods and clearly show the impact of their architectural modifications on model performance. This is one of the clearest papers I have read.

**Weaknesses:**

- The primary weakness of this paper is that their VAE model does not outperform existing unsupervised variant effect predictors (like ESM or DeepSequence). They find that using a more complicated prior does not improve performance and that self-attention layers with attention maps defined using AF2 contacts does not help.
- The authors do go on to say that their encoder-only model trained on DMS data does outperform unsupervised variant effect predictors, but they do not compare to unsupervised variant effect predictors fine-tuned on DMS data. Their are many ways this fine-tuning has been proposed in the past and the authors should benchmark matENC-DMS to those methods: https://pubmed.ncbi.nlm.nih.gov/35039677/, https://www.nature.com/articles/s41467-024-51844-2, and https://arxiv.org/abs/2405.06729.
- The authors should try pre-training a VAE on MSA data and then fine-tuning the encoder of the MSA on DMS data.
- A central goal of the paper seems to be to identify the settings in which DMS data is useful for improving variant effect predictions. In Fig. A8, the authors are unable to find any correlations between metadata of the protein and performance difference between the DMS-trained and MSA-trained models. However, they don't consider structural features of the protein itself. Analysis along those lines would be interesting.

**Questions:**

- How much does performance change if attention maps in self-attention layers are learned as opposed to derived from AF2 predicted contacts?

- Based on my understanding of Fig. 1, the hyperparameter $d$ should be 20, corresponding to the number of amino acids. Why do you not embed amino acids in a higher dimensional space as you get deeper into the VAE?

- Instead of using a dimension-wise FC layer, do you think it would be valuable to have an attention pooling layer to reduce the number of parameters (# of parameters in attention pooling does not scale with the length of the sequence)? It also automatically allows you to handle sequences of different lengths.

- For matENC-DMS, are variants at the same position either all in the training set or the testing set? Training on some varaints and testing on other variants at the same position has been shown to be a form of data leakage that inflates performance.

- Does difference in performance between matENC-DMS and matVAE-MSA depend on the number of variants assayed in the DMS? Could you include that In Fig. A8?

---

> ### Author Response · Authors · 2024-11-20
> **Authors response 1/2**
>
> *Weaknesses:*\
> *The primary weakness of this paper is that their VAE model does not outperform existing unsupervised variant effect predictors (like ESM or DeepSequence).*\
> ◦ Thank you for this comment. Although our architectures do not outperform most baselines models on average, we found that our zero-shot prediction model outperforms other methods for 2 proteins, both related to the Noonan syndrome. We now state this more clearly in the results section as well as in the abstract.
>
> *They find that using a more complicated prior does not improve performance and that self-attention layers with attention maps defined using AF2 contacts does not help.
> The authors do go on to say that their encoder-only model trained on DMS data does outperform unsupervised variant effect predictors, but they do not compare to unsupervised variant effect predictors fine-tuned on DMS data. Their are many ways this fine-tuning has been proposed in the past and the authors should benchmark matENC-DMS to those methods: https://pubmed.ncbi.nlm.nih.gov/35039677/, https://www.nature.com/articles/s41467-024-51844-2, and https://arxiv.org/abs/2405.06729.*\
> ◦ Thank you for this observation. We now provide a comparison with supervised learning approach reported in ProteinGym in Table 2.a.  These methods are fine tuning of models trained on MSA and therefore the comparison with our methods is not exactly fair, but we acknowledge that the comparison is a very valuable addition to the paper.
>
> *The authors should try pre-training a VAE on MSA data and then fine-tuning the encoder of the MSA on DMS data.*\
> ◦ We thank the reviewer for suggesting this analysis. We do acknowledge that fine tuning might improve performances. Nonetheless the primary goal of the paper is to evaluate the evolutionary assumption and we do this by training models of similar capacity from scratch on either MSA or DMS datasets. We argue that fine tuning would not give further insights on the evolutionary assumption. This is nonetheless an important future work that we now mention in the Future work section.
>
> *A central goal of the paper seems to be to identify the settings in which DMS data is useful for improving variant effect predictions. In Fig. A8, the authors are unable to find any correlations between metadata of the protein and performance difference between the DMS-trained and MSA-trained models. However, they don't consider structural features of the protein itself. Analysis along those lines would be interesting.*\
> ◦ Thank you for this suggestion. We found that analysis considering details structural features of protein might be out of scope for a machine learning conference. We therefore limited ourselves to dataset information and a few characteristics that are classical in pharmacogenomics.

---

> > ### Author Response · Authors · 2024-11-20
> > **Authors response 2/2**
> >
> > *Questions:*\
> >
> > *How much does performance change if attention maps in self-attention layers are learned as opposed to derived from AF2 predicted contacts?*\
> > ◦ Thank you for this suggestion. This is an interesting analysis that we are currently performing. We prefer to proceed with the rebuttal and hope to be able to include it in a latter revision of the paper.
> >
> > *Based on my understanding of Fig. 1, the hyperparameter “d” should be 20, corresponding to the number of amino acids. Why do you not embed amino acids in a higher dimensional space as you get deeper into the VAE? Instead of using a dimension-wise FC layer, do you think it would be valuable to have an attention pooling layer to reduce the number of parameters (# of parameters in attention pooling does not scale with the length of the sequence)? It also automatically allows you to handle sequences of different lengths.*\
> > ◦ Thank you for these design suggestions. One of our motivation for our design was to improve the input layer in deepsequence. In deepsequence the authors use the flattened 20d one-hot codes (or 1x1 convolution of one-hot codes, still resulting in 20d embeddings) as input to their fully-connected MLP. In our case we  compare the embeddings in deepsequence with embeddings learnt with a transformer, which is why we fixed the dimension to 20. Regarding the pooling layer, we share weights across dimension to reduce the length dimension of the input sequence. This can be seen as a trainable pooling layer. We did not consider more advanced pooling since we were interested in the analogy with linear decomposition methods, this is however an interesting avenue for further research.
> >
> > *For matENC-DMS, are variants at the same position either all in the training set or the testing set? Training on some varaints and testing on other variants at the same position has been shown to be a form of data leakage that inflates performance.*\
> > ◦ We acknowledge that the same position may be used for training and testing, however the problem also exists when training on MSA and testing on DMS, since the position of the variants are not controlled when training on MSA in the literature. Our approach for cross-validation of matENC-DMS is similar to other works, e.g. Gitter et al, and is also one of the methods used in ProteinGym.
> >
> > *Does difference in performances between matENC-DMS and matVAE-MSA depend on the number of variants assayed in the DMS? Could you include that In Fig. A8?*\
> > ◦ The graph has been added to the correlation study in figure A.9.  Visually the plot shows a more convincing correlation than the others, but the correlation is still not statistically significant. We nonetheless thank the reviewer because this probably points to a limit of our method of only using DMS data for training.

---

> > > ### Comment · Reviewer_y9E7 · 2024-11-21
> > > **Response to authors**
> > >
> > > Thank you for including comparisons to unsupervised predictors fine-tuned on DMS data, as well as the other clarifications. Given that matVAE-MSA does not on average outperform other unsupervised predictors  and matENC-DMS does not on average outperform other supervised predictors, the architectural improvements tried in the paper do not seem to be particularly valuable.
> > >
> > > I do appreciate the authors' desire to better understand when DMS datasets improve variant effect predictions. However, given that the authors do not find features that strongly correlate with the difference in DMS v MSA performance and given that it has been proven numerous times before that fine-tuning on DMS data is helpful, I would like to keep my score.

---

### Meta-Review · Area_Chair_Pk7A · 2024-12-19

**Metareview:**

The paper considers the problem of variant effect prediction and proposes a  transformer-based matrix VAE architecture to tackle this problem (matVAE-MSA ). This approach is compared to matENC-DMS, a model with similar capacity that is leveraging DMS data on 33 DMS datasets from ProteinGym.

Unfortunately, the proposed model does not provide significant advantages compared to existing approaches. Moreover the authors do not provide conclusive new insights on settings where using DMS data is valuable.

**Additional Comments On Reviewer Discussion:**

The key concerns discussed included  the lack of significant improvement on performance, lack of novelty of the methodological contributions and the lack of novel insights.

These concerns were shared by all reviewers, and unfortunately none of these concerns were satisfactorily addressed during rebuttal as summarized by Reviewer y9E7.

Moreover as noted by Reviewer QCTd the additional supervised learning experiment provided by the authors in response to the reviewer concerns further confirm the lack of significant improvement of the proposed approach.

---

### Decision · Program_Chairs · 2025-01-22

Reject